# Emergent dynamics of adult stem cell lineages from single nucleus and single cell RNA-Seq of *Drosophila* testes

Amelie A Raz[1†], Gabriela S Vida[2†], Sarah R Stern[3†], Sharvani Mahadevaraju[4†], Jaclyn M Fingerhut[1†], Jennifer M Viveiros[5†], Soumitra Pal[6†], Jasmine R Grey[5†], Mara R Grace[5†], Cameron W Berry[3], Hongjie Li[7], Jasper Janssens[8], Wouter Saelens[9], Zhantao Shao[10], Chun Hu[10], Yukiko M Yamashita[1], Teresa Przytycka[6], Brian Oliver[4], Julie A Brill[11,12,13], Henry Krause[10,12], Erika L Matunis[5], Helen White-Cooper[14], Stephen DiNardo[2*], Margaret T Fuller[3,15*]

[1]Whitehead Institute for Biomedical Research and Department of Biology, Massachusetts Institute of Technology, Howard Hughes Medical Institute, Cambridge, United States; [2]Department of Cell and Developmental Biology, The Perelman School of Medicine and The Penn Institute for Regenerative Medicine, Philadelphia, United States; [3]Department of Developmental Biology, Stanford University School of Medicine, Stanford, United States; [4]National Institute of Diabetes and Digestive and Kidney Diseases, National Institutes of Health, Bethesda, United States; [5]Department of Cell Biology, Johns Hopkins University School of Medicine, Baltimore, United States; [6]National Center for Biotechnology Information, National Library of Medicine, National Institutes of Health, Bethesda, United States; [7]Huffington Center on Aging and Department of Molecular and Human Genetics, Baylor College of Medicine, Houston, United States; [8]JVIB Center for Brain & Disease Research, and the Department of Human Genetics, KU Leuven, Leuven, Belgium; [9]Data Mining and Modeling for Biomedicine, VIB Center for Inflammation Research, and Department of Applied Mathematics, Computer Science and Statistics, Ghent University, Ghent, Belgium; [10]Donnelly Centre for Cellular and Biomolecular Research, University of Toronto, Toronto, Canada; [11]Cell Biology Program, The Hospital for Sick Children, Toronto, Canada; [12]Department of Molecular Genetics, University of Toronto, Toronto, Canada; [13]Institute of Medical Science, University of Toronto, Toronto, Canada; [14]School of Biosciences, Cardiff University, Cardiff, United Kingdom; [15]Department of Genetics, Stanford University, Stanford, United States

*For correspondence:
sdinardo@pennmedicine.upenn.edu (SDiN);
mtfuller@stanford.edu (MTF)

†These authors contributed equally to this work

**Abstract** Proper differentiation of sperm from germline stem cells, essential for production of the next generation, requires dramatic changes in gene expression that drive remodeling of almost all cellular components, from chromatin to organelles to cell shape itself. Here, we provide a single nucleus and single cell RNA-seq resource covering all of spermatogenesis in *Drosophila* starting from in-depth analysis of adult testis single nucleus RNA-seq (snRNA-seq) data from the Fly Cell Atlas (FCA) study. With over 44,000 nuclei and 6000 cells analyzed, the data provide identification of rare cell types, mapping of intermediate steps in differentiation, and the potential to identify new factors impacting fertility or controlling differentiation of germline and supporting somatic cells. We justify assignment of key germline and somatic cell types using combinations of known markers, in situ hybridization, and analysis of extant protein traps. Comparison of single cell and single nucleus datasets proved particularly revealing of dynamic developmental transitions in germline

differentiation. To complement the web-based portals for data analysis hosted by the FCA, we provide datasets compatible with commonly used software such as Seurat and Monocle. The foundation provided here will enable communities studying spermatogenesis to interrogate the datasets to identify candidate genes to test for function in vivo.

## Editor's evaluation

This paper uses single-nucleus RNA-seq and new single-cell RNA-seq data to provide an extensive characterization of cell types found within the *Drosophila* testis. This work provides a detailed analysis of the developmental transitions experienced by cells in the germline stem cell and cyst stem cell lineages. Comparison of snRNA-seq and scRNA-seq also reveals differences in steady-state RNA levels and ongoing transcription, providing insights into how gene expression patterns change during the development of these different cell populations. This study provides an important roadmap for future work using *Drosophila* testes to study development, reproduction, and stem cell biology.

## Introduction

Single cell RNA-seq (scRNA-seq) of developing tissues can reveal new cell types as well as previously unknown steps in the differentiation of lineages underlying tissue homeostasis and repair. In fact, high-resolution expression maps are being created for entire organisms, from *C. elegans*, planaria, and schistosomes to *Drosophila* and mouse (*Cao et al., 2017*; *Cao et al., 2019*; *Fincher et al., 2018*; *Li et al., 2022*; *Plass et al., 2018*; *Schaum et al., 2018*; *Sebé-Pedrós et al., 2018*; *Siebert et al., 2019*; *Wendt et al., 2020*), with such atlases providing a foundational reference for several important model organisms. In particular, for tissues maintained by stem cell lineages, scRNA-seq can identify the developmental trajectories that lead from dedicated tissue stem cell to terminally differentiated cell types, an important resource for understanding tissue maintenance, repair, and the origins of cancer.

The testis harbors a highly active, unipotent adult stem cell lineage that must produce sperm throughout reproductive life. Spermatogenesis relies on self-renewing germline stem cells, the progeny of which differentiate into one of the most highly specialized cell types in the body. Production of functional sperm requires intimate interactions between germ cells and somatic support cells, with defects at almost any step compromising fertility. Interest in spermatogenesis has motivated scRNA-seq analyses of testes from a variety of organisms, including mouse (*Cao et al., 2021*; *Chen et al., 2018*; *Green et al., 2018*; *Guo et al., 2020*; *Law et al., 2019*) and *Drosophila* (*Li et al., 2022*; *Mahadevaraju et al., 2021*; *Witt et al., 2019*). Notably, the testis of *Drosophila* has the highest complexity in terms of mRNAs expressed of any tissue in the fly, likely reflecting the dramatic differentiation events required (*Li et al., 2022*).

Many aspects of spermatogenesis are conserved from *Drosophila* to mammals. One striking difference, however, is that spermatogenesis in *Drosophila* relies on not one but two adult stem cell lineages. The co-differentiating germ cells and their closely associated somatic support cells descend from distinct stem cell populations, housed together in a well-defined niche (*Fuller, 1998*). Additionally, the many mutations affecting male fertility, plus powerful genetic tools for cell-type-specific functional analysis, have allowed identification of stage-specific regulatory factors underlying niche function in stem cell maintenance, control of proliferation, and soma-germline feedback circuits that act during co-differentiation of these two lineages. This comprehensive knowledge of spermatogenesis offers a rich biological foundation for interpreting single nucleus and single cell RNA-seq data.

Here, we present an in-depth analysis of the testis subset of the Fly Cell Atlas (FCA) single nucleus RNA-Seq (snRNA-seq) data. We supplement this with scRNA-seq from the same tissue, together providing a foundational reference for the field. While several recent RNA-seq analyses of *Drosophila* testes have been illuminating, they generally focused on particular stages (*Gan et al., 2010*; *Hof-Michel and Bökel, 2020*; *Lu et al., 2020*; *Mahadevaraju et al., 2021*; *Shi et al., 2020*; *Vedelek et al., 2018*; *Witt et al., 2019*). In contrast, the scale and comprehensive nature of the FCA dataset allowed us to profile rare cell types, such as the stem cell niche, and to follow spermatogenesis from early spermatogonia to late spermatids, a remarkable conversion of precursors to highly elongated, specialized cells. We present supporting data for assignment of key cell types, both germline and

somatic, and show how progression through two distinct, yet intimately interacting, stem-cell-based lineages emerges from the changes in gene expression.

The data confirm and extend known features of the male germ line transcription program, including cell-type-specific expression of many genes in spermatocytes, downregulation of X-linked genes in later spermatocytes, and repression of most transcription in early spermatids. At the same time, surprising new features emerged, including unexpected complexity in the somatic support cell lineage. In addition, comparison of single nucleus with single cell sequencing data significantly expanded our understanding of gene expression dynamics in spermatocytes and spermatids as they mature. In particular, these data showed how dynamic changes in active transcription reflected in the snRNA-seq can be obscured by the endowment of mRNAs stored in the cytoplasm. This is especially clear in early haploid spermatids, which have little transcriptional activity but contain many mRNAs transcribed in spermatocytes and stored to be translated later to support spermatid morphogenesis. With a gene expression framework for the two testis stem cell lineages now in place, mining the snRNA-seq data for changes in gene expression as one cluster advances to the next should identify new sub-stage-specific markers, thereby opening the way for tests of function for such newly identified genes in male germ cell differentiation.

## Results

### Clustering by gene expression signature reveals progression of differentiation in two stem cell lineages

Spermatogenesis in *Drosophila* involves obligate, intimate interactions between cells differentiating in two adult stem cell-founded lineages. Male germline stem cells (GSCs) and their partners, the somatic cyst stem cells (CySCs), are both physically anchored to a small cluster of somatic cells termed the apical hub (*Figure 1A*), which provides short-range niche signals important for maintenance of the two stem cell states. The interleaved arrangement of GSCs and CySCs ensures that their immediate daughters are positioned to interact. Two postmitotic early cyst cells enclose each gonialblast (immediate GSC daughter), forming a two-cell squamous epithelium that later seals the progeny of the gonialblast off from the rest of the testis (*Fairchild et al., 2015*). The gonialblast initiates four rounds of spermatogonial transit amplifying divisions with incomplete cytokinesis, producing 16 interconnected germ cells (*Figure 1A*). After the fourth mitosis, the germ cells undergo premeiotic S-phase and enter an extended G2 cell cycle phase termed meiotic prophase. Over the next three and a half days the 16 primary spermatocytes increase 25-fold in volume, engaging in a robust transcription program in preparation for the meiotic divisions and the extensive elongation and remodeling of the resulting 64 haploid spermatids into mature sperm. Although they do not divide, the two somatic cyst cells co-differentiate with the germ cells they enclose (*Gönczy et al., 1992*), eventually taking on different identities as head and tail cyst cells. The head cyst cell cups the nuclear end of elongating spermatid bundles and eventually inserts into the terminal epithelium at the base of the testis, while the tail cyst cell elongates extensively to cover the rest of the spermatid bundle (*Tokuyasu et al., 1972*). All these cell types, as well as somatic structural cells of the testis sheath (muscle and pigment cells) and cells of the seminal vesicle are represented in the FCA testis dataset.

The relative similarity and differences in gene expression for 44,621 single nuclei from triplicate 10 X snRNA-seq runs from adult testis plus seminal vesicle (see Materials and methods) can be visualized in a Uniform Manifold Approximation and Projection (UMAP)-based dimensionality reduction plot (*Figure 1B*). The geography of the UMAP is dominated by the dynamic sequences of differentiating states in the germline (blue) and somatic cyst cell (yellow) lineages. Each lineage manifests as an emergent trajectory of nuclei with continuously progressing gene expression profiles, unlike the discrete clusters characteristic of most terminally differentiated cell types. Despite their physical proximity and cooperation in vivo (blue and yellow in *Figure 1A*), the germ line and cyst cell lineages mapped to largely non-overlapping formations in gene expression space represented in the UMAP (blue and yellow in *Figure 1B*), consistent with their different embryological origin, cell biology, and known roles.

From the perspective in *Figure 1B*, the spatial arrangement of nuclei in the UMAP whimsically resembles a hammerhead shark (blue - germ line) playing a saxophone (yellow - cyst cell lineage) watched over by a mermaid (several somatic epithelial-based structural elements, including the

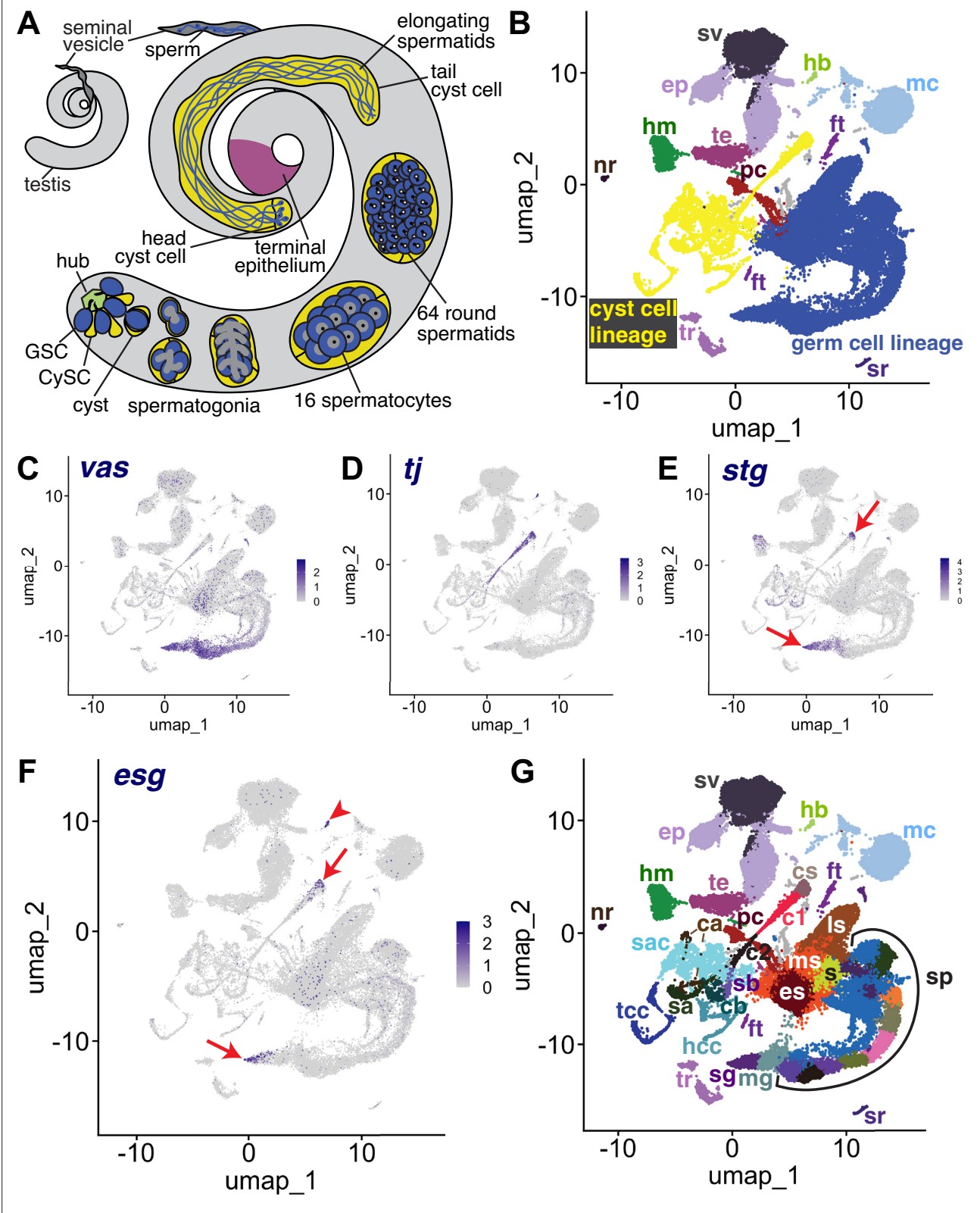

**Figure 1.** The snRNA-seq landscape of the testis. (**A**) Illustration of adult *Drosophila* testis showing hub (green), germ cell lineage (blue), cyst cell lineage (yellow), terminal epithelium (pink), and seminal vesicle (dark gray). (**B**) UMAP of FCA snRNAseq data from the testis plus seminal vesicle (relaxed version). Blue: germ cell lineage; Yellow: cyst cell lineage; Pink: terminal epithelial cells of testis (te); Dark gray: seminal vesicle (sv). Other cell types as listed in (**G**). (**C–F**) UMAP plots of snRNA-seq data showing expression of: (**C**) *vasa* (*vas*), (**D**) traffic *jam* (*tj*), (**E**) *string* (*stg*), (**F**) *escargot* (*esg*). Red arrows:

*Figure 1 continued on next page*

*Figure 1 continued*

proliferating cells. Red arrowhead: hub. (**G**) UMAP (as in **B**) with Leiden 6.0 clusters of germ and cyst cell lineages labeled (sg: Spermatogonia; mg: Mid-late proliferating spermatogonia; sp: Spermatocytes; s: Spermatids; es: Early elongation-stage spermatids; ms: Early-mid elongation stage spermatids; ls: Mid-late elongation-stage spermatids; hb: Germinal proliferation center hub; cs: Cyst stem cells, c1: Early cyst cells 1; c2: Early cyst cells 2; sa: Cyst cell with spermatocytes branch A; sb: Cyst cell with spermatocytes branch B; ca: Cyst cell branch a; cb: Cyst cell branch b; sac: Elongating spermatid-associated cyst cells; hcc: Head cyst cells; tcc: Tail cyst cells; te: Terminal epithelial cells of testis; sv: Seminal vesicle; ep: Male gonad associated epithelium; sr: Secretory cells of the male reproductive tract; mc: Muscle cells; hm: Hemocytes; nr: Neurons; pc: Pigment cells;, tr: Trachea; ft: Fat body).

The online version of this article includes the following figure supplement(s) for figure 1:

**Figure supplement 1.** Expression of *vasa* in the snRNA-seq dataset.

seminal vesicle (sv, dark gray) and terminal epithelial cells at the testis base (te, pink)). One notable cluster located near the mermaid head is the hub (hb, light green), the niche that supports the two stem cell lineages. Other clusters on the UMAP contain differentiated cell types that contribute to organ structure, including muscle (mc) and pigment cells (pc) of the testis sheath (*Figure 1B*). Additionally, sample dissection carried over small numbers of non-testis cells, including tracheal (tr) and fat body (ft) cells, hemocytes (hm), neurons (nr), and male reproductive tract secretory cells (sr).

Identity of key clusters was assigned based on expression of known markers from the literature (citations for all published markers employed given in *Table 1*). Expression of *vasa (vas)* identified early germ line nuclei while expression of *traffic jam (tj)* identified nuclei from early stages in the somatic cyst cell lineage (*Figure 1C and D*). A small proportion of nuclei in other tissues reported *vas* expression at a low level, perhaps due to ambient RNA in the nuclear preparation resulting from lysed cells rather than reflecting *vas* expression outside the germ line (*Figure 1—figure supplement 1A*). However, it is also worth noting that Vasa has been shown to be expressed in somatic cells of the gonad (*Renault, 2012*). Expression of the cdc25 phosphatase *string (stg)*, required for the G2/M transition in mitotic cells (*Alphey et al., 1992*; *Edgar and O'Farrell, 1990*), and *escargot (esg)*, a gene expressed in diploid proliferative cells (*Fuse et al., 1994*), marked CySCs in the cyst cell lineage and proliferating GSCs and spermatogonia in the germ line lineage (*Figure 1E and F*). Expression of *esg* also marked the hub, as expected from prior studies (*Voog et al., 2014*; *Figure 1F*). Together, these markers established that the germ line lineage begins at the tail end of the 'shark' with germ line stem cells (GSCs) and proliferating spermatogonia at the tapered point. The somatic cyst cell lineage begins at the mouthpiece of the 'saxophone' at the UMAP center, with early cyst cell nuclei extending down and leftward in a thin line. In addition, analysis by fluorescent in situ hybridization (FISH), and the average number of unique transcripts (Unique Molecular Identifier - UMI) expressed helped assign identity. For example, spermatocytes are highly transcriptionally active, whereas *Drosophila* early spermatids are nearly quiescent. While clustering was carried out using the Leiden algorithm at increasing levels of resolution, we settled on Leiden 6.0 as providing optimal granularity along both somatic and germ-line differentiation trajectories. We assigned 43 clusters as germline and 22 clusters as likely cyst cell lineage, with many inferred from the UMAP geography as representing putative intermediate cell types in the respective lineages (*Figure 1G*, *Figure 2A*, Figure 6A and *Figure 2—figure supplement 1A*).

## Progression of differentiation in the male germ line stem cell lineage

*Figure 2A* shows the UMAP for the germ line stem cell lineage with Leiden 6.0 clusters labeled. Expression of the germ-cell-specific gap junction gene *zero population growth (zpg)*, required for survival of early spermatogonia (*Tazuke et al., 2002*; *Figure 2D, L*), along with *vasa, stg* and *esg* (*Figure 1C, E, F*), further established nuclei at the pointed tip of the shark tail (clusters 25 and 22) as GSCs and spermatogonia. In vivo, GSCs are distinguished from gonialblasts and transit amplifying spermatogonia cytologically, by attachment to the apical hub and cell biological characteristics such as oriented centrosomes and spindles, and functionally, by lineage analysis. However, mRNA markers restricted to GSCs have not yet been identified, preventing us from determining what percent of these early nuclei are GSCs. Many nuclei in cluster 22 express *bag-of-marbles (bam)* (*Figure 2L*) but lack known spermatocyte markers, suggesting that these nuclei represent mid-to-late spermatogonia or germ cells undergoing premeiotic S phase. Moving rightward, several known early spermatocyte markers such as *kumgang (kmg)* and *RNA-binding protein 4 (Rbp4)* began to be expressed (clusters 5, 78, and 40) (*Figure 2F–G, L*). Transcripts from *aubergine (aub),* a piRNA binding protein, were detected in the

**Table 1.** List of identifying genes.

| Gene symbol | Gene_name | FBgn | Reference | DOI |
|---|---|---|---|---|
| aly | Always early | FBgn0004372 | *White-Cooper et al., 2000* | 10.1242/dev.127.24.5463 |
| aub | Aubergine | FBgn0000146 | *Nishida et al., 2007* | 10.1261/rna.744307 |
| bam | Bag of marbles | FBgn0000158 | *Schulz et al., 2004* | 10.1534/genetics.103.023184 |
| CadN | Cadherin-N | FBgn0015609 | *Boyle et al., 2007* | 10.1016 /j.stem.2007.08.002 |
| can | Cannonball | FBgn0011569 | *Hiller et al., 2001* | 10.1101/gad.869101 |
| cher | Cheerio | FBgn0014141 | *Tanentzapf et al., 2007* | 10.1038/ncb1660 |
| CycB | Cyclin B | FBgn0000405 | *White-Cooper et al., 1998* | 10.1242/dev.125.1.125 |
| Dic61B | Dynein intermediate chain at 61B | FBgn0263988 | *Lu et al., 2020* | 10.1101/gad.335331.119 |
| dlg1 | Discs large 1 | FBgn0001624 | *Papagiannouli and Mechler, 2009* | 10.1038/cr.2009.71 |
| esg | Escargot | FBgn0287768 | *Kiger et al., 2001* | 10.1038/35037606 |
| eya | Eyes absent | FBgn0000320 | *Fabrizio et al., 2003* | 10.1016/s0012-1606(03)00127–1 |
| f-cup | Flyers-cup | FBgn0028487 | *Barreau et al., 2008* | 10.1242/dev.021949 |
| Fas3 | Fasciclin III | FBgn0000636 | *Brower et al., 1981* | 10.1242/dev.63.1.233 |
| fzo | Fuzzy onions | FBgn0011596 | *Hwa et al., 2002* | 10.1016/s0925-4773(02)00141–7 |
| hh | Hedgehog | FBgn0004644 | *Michel et al., 2012* | 10.1242/dev.075242 |
| Hml | Hemolectin | FBgn0029167 | *Li et al., 2022* | 10.1126/science.abk2432 |
| kl-2 | Male fertility factor kl2 | FBgn0001313 | *Carvalho et al., 2000* | 10.1073/pnas.230438397 |
| kl-3 | Male fertility factor kl3 | FBgn0267432 | *Carvalho et al., 2000* | 10.1073/pnas.230438397 |
| kl-3 | Male fertility factor kl3 | FBgn0267432 | *Fingerhut et al., 2019* | 10.1371/journal.pgen.1008028 |
| kl-5 | Male fertility factor kl5 | FBgn0267433 | *Gepner and Hays, 1993* | 10.1073/pnas.90.23.11132 |
| kl-5 | Male fertility factor kl5 | FBgn0267433 | *Fingerhut et al., 2019* | 10.1371/journal.pgen.1008028 |
| kmg | Kumgang | FBgn0032473 | *Kim et al., 2017* | 10.1126/science.aal3096 |
| Mst77F | Male-specific transcript 77 F | FBgn0086915 | *Barckmann et al., 2013* | 10.1016 /j.ydbio.2013.02.018 |
| Mst84Db | Male-specific RNA 84 Db | FBgn0004173 | *Kuhn et al., 1991* | 10.1016/0925-4773(91)90064-d |
| Mst84Dc | Male-specific RNA 84Dc | FBgn0004174 | *Kuhn et al., 1991* | 10.1016/0925-4773(91)90064-d |
| Mst87F | Male-specific RNA 87 F | FBgn0002862 | *Kuhn et al., 1991* | 10.1016/0925-4773(91)90064-d |
| MtnA | Metallothionein A | FBgn0002868 | *Gan et al., 2010* | 10.1093/nar/gkp1006 |
| Nep5 | Neprilysin 5 | FBgn0039478 | *Sitnik et al., 2014* | 10.1534/genetics.113.160945 |
| p53 | p53 | FBgn0039044 | *Monk et al., 2012* | 10.1007/s00441-012-1479-4 |
| p-cup | Presidents-cup | FBgn0030840 | *Barreau et al., 2008* | 10.1242/dev.021949 |
| piwi | P-element induced wimpy testis | FBgn0004872 | *Gonzalez et al., 2015* | 10.1016 /j.celrep.2015.06.004 |
| Rbp4 | RNA-binding protein 4 | FBgn0010258 | *Baker et al., 2015* | 10.1242/dev.122341 |
| sa | Spermatocyte arrest | FBgn0002842 | *Hwa et al., 2004* | 10.1242/dev.01314 |
| shg | Shotgun | FBgn0003391 | *Voog et al., 2008* | 10.1038/nature07173 |
| so | Sine oculis | FBgn0003460 | *Fabrizio et al., 2003* | 10.1016/s0012-1606(03)00127–1 |
| soti | Scotti | FBgn0038225 | *Barreau et al., 2008* | 10.1242/dev.021949 |
| stg | String | FBgn0003525 | *Alphey et al., 1992* | 10.1016/0092-8674(92)90616-k |
| Syt1 | Synaptotagmin 1 | FBgn0004242 | *Li et al., 2022* | 10.1126/science.abk2432 |
| tj | Traffic jam | FBgn0000964 | *Li et al., 2003* | 10.1038/ncb1058 |

*Table 1 continued on next page*

*Table 1 continued*

| Gene symbol | Gene_name | FBgn | Reference | DOI |
|---|---|---|---|---|
| tomboy20 | Tomboy20 | FBgn0037828 | *Hwa et al., 2004* | 10.1016 /j.febslet.2004.07.025 |
| upd1 | Unpaired 1 | FBgn0004956 | *Tulina and Matunis, 2001* | 10.1126/science.1066700 |
| vas | Vasa | FBgn0283442 | *Hay et al., 1988* | 10.1016/0092-8674(88)90216-4 |
| wa-cup | Walker cup | FBgn0037502 | *Barreau et al., 2008* | 10.1242/dev.021949 |
| zpg | Zero population growth | FBgn0024177 | *Tazuke et al., 2002* | 10.1242/dev.129.10.2529 |

spermatogonial region (clusters 25, 22) and overlapping with early spermatocyte markers (clusters 5, 78, 40, and 41) (*Figure 2E and L*). Fluorescent in situ hybridization (FISH) confirmed *aub* transcripts present in GSCs around the hub, spermatogonia, and extending into early spermatocyte cysts, with their characteristic larger nuclei (*Figure 2B*). FISH also confirmed expression of *kmg* mRNA starting in early spermatocytes (*Figure 2C*), with early spermatocytes showing both *aub* and *kmg* transcripts, consistent with the snRNA-seq data.

Progressively maturing spermatocytes along the bottom right of the germline UMAP expressed later markers, including mRNAs for the spermatocyte-specific tMAC subunit *always early* (*aly*) and the testis-specific TAFs (tTAFs) *spermatocyte arrest (sa)* and *cannonball (can)* (clusters 41, 51, 35, and onward; *Figure 2H and L*). Expression of *fuzzy onions* (*fzo*) and *Dynein intermediate chain 61B* (*Dic61B*) was detected later, as the germ cell clusters curved upward (clusters 33, 48,105, 45, 13, 56; *Figure 2J and L*), consistent with the dependence of *fzo* and *Dic61B* transcription on *aly* (*Hwa et al., 2002*; *Lu et al., 2020*). Correlating in vivo morphology with gene expression space (visualized in the UMAP), *fzo* transcripts were not detected by FISH in the young spermatocytes near the spermatogonial region but were strongly detected in more mature spermatocytes further away from the testis apical tip (*Figure 2B*). The G2/M cell cycle regulator *CyclinB* (*CycB*) is transcribed from one promoter in mitotic spermatogonia, silenced, then re-expressed from an alternate promoter in later spermatocytes, dependent on *aly* function (*Lu et al., 2020*; *White-Cooper et al., 1998*). These two distinct stages of *CycB* transcript expression are clearly visible in the snRNA-seq data (*Figure 2I and L*). Maturing spermatocytes, marked by expression of Y-linked genes (*Figure 3D*), lie toward the top of the upward curve where the tail meets the torso of the shark. The progression of germ cell differentiation continues with early stage spermatids along the upper torso and head of the shark (marked by low UMI - see below). Mid-to-late elongation stage spermatids, marked by expression of *p-cup* mRNA, lie in the blunt projection toward the upper right of the UMAP (clusters 66, 10, 46; *Figure 2K and L*).

The order of clusters in expression space reflects differentiation in the lineage, as indicated by plotting the expression of known germline markers in each UMAP cluster (*Figure 2L*). Notably, using the published marker genes scored here, sequential cluster identities (e.g. 25, 22, 5, 78, 40) were not each delimited by unique marker genes. Instead, graded expression of the markers examined extended across boundaries between clusters, as expected for a continuous differentiation process. This was also observed in a UMAP with just nine clusters created at lower resolution of Leiden 0.8 (*Figure 2—figure supplement 1B and C*). In principle, there may also be genes where expression is delimited by cluster boundaries, and these may identify novel markers demarcating germline transition stages. Four genes were selected to test using RNA FISH as potentially meeting this criterion. Two represented the earliest cluster (GSCs and early spermatogonia, *Pxt* and *Drep2*) and two represented a mid-trajectory cluster (mid-spermatocytes, *CG43317* and *Vha16-5*). As predicted, *Pxt* and *Drep2 expression was* restricted to early germline stages, while *CG43317* and *Vha16-5* were robustly expressed in spermatocytes and not detectable at early stages (*Figure 2—figure supplement 1D–E*, genes marked with solid squares in *Figure 4G–H*). The results validate use of these genes as new markers for their respective stages, and demonstrate the general utility of the dataset for discovery of novel markers.

The geography of the UMAP is reminiscent of the spatio-temporal organization in the testis itself, with stages laid out from GSCs through transit amplifying spermatogonia, and then through young, mid, and late spermatocyte stages. However, a UMAP displays changes in gene expression space

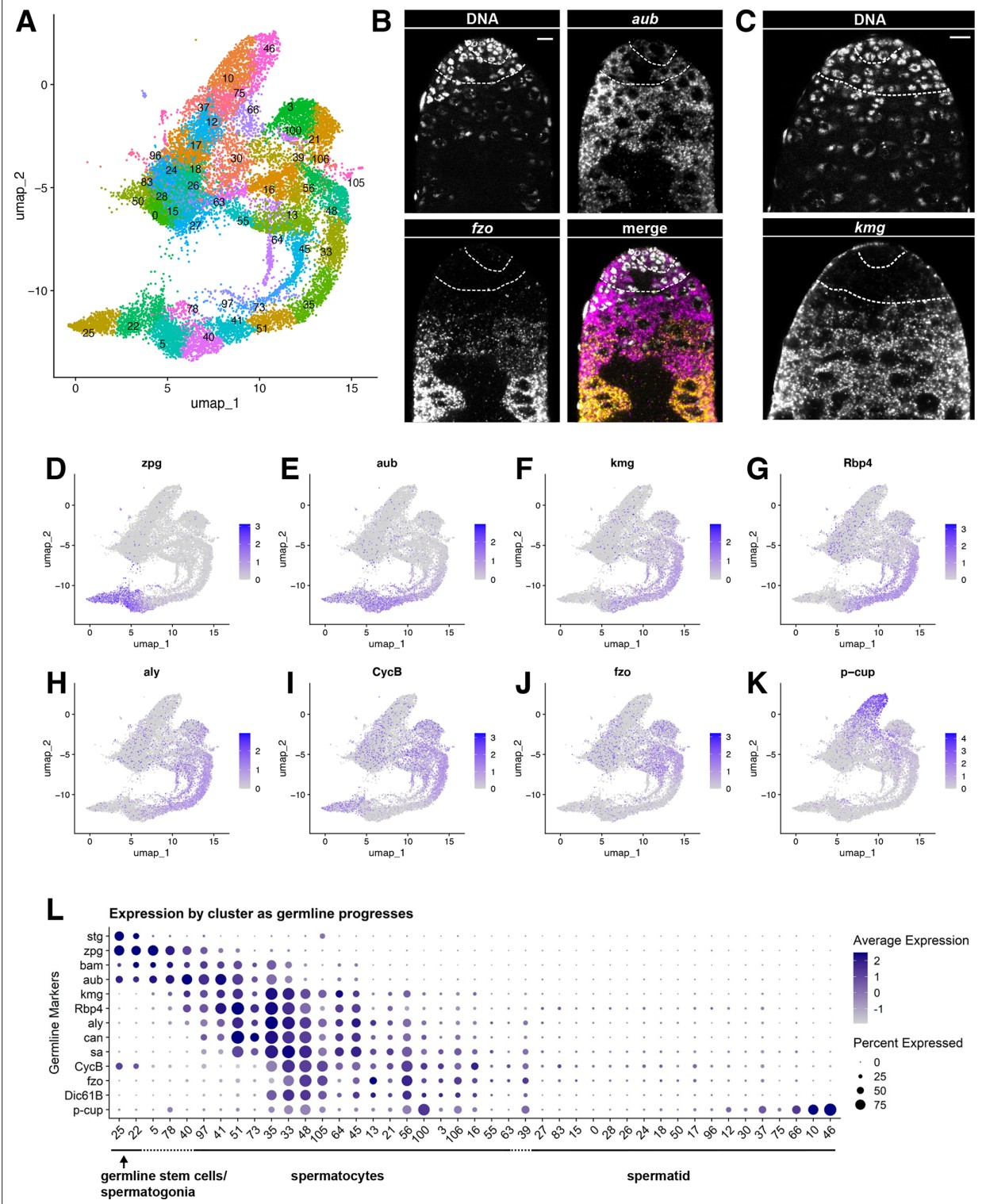

**Figure 2.** Characteristics of the germline lineage. (**A**) Germline portion of the UMAP generated by Seurat from clustering of the full testis plus seminal vesicle dataset at Leiden 6.0 resolution. (**B–C**) Apical tips of testes showing localized expression of (**B**) *aub* (magenta) and *fzo* (yellow) mRNA and (**C**) *kmg* mRNA visualized by in situ hybridization. Apical-most dotted line demarcates germ line stem cells (GSCs) around the hub from spermatogonia. Lower dotted line demarcates spermatogonia and cells in premeiotic S phase from young spermatocytes. Scale bars, 10 μm. (**D–K**) Feature plots generated by Seurat showing expression levels of *zpg, aub, kmg, rbp4, aly, CycB, fzo,* and *p-cup* in the germline UMAP. Navy blue gradient bars: relative expression level for the indicated gene. (**L**) Dot plot generated by Seurat showing expression levels of selected germline markers by cluster as nuclei progress from

*Figure 2 continued on next page*

*Figure 2 continued*

spermatogonia to spermatid. Color intensity: level of expression of the indicated gene averaged over all the nuclei in a given cluster relative to the level for that gene in other germline clusters. Size of dots: percent of nuclei in specified cluster in which expression of the gene was detected (see also *Figure 2—figure supplement 1*).

The online version of this article includes the following figure supplement(s) for figure 2:

**Figure supplement 1.** Comparison of snRNA-seq data clustered at Leiden 6.0 vs. lower resolution and new marker genes.

rather than physical space. As a consequence, some surprisingly long stretches of the germline testis UMAP represent what are known to be short periods in developmental time. For example, the short, young spermatocyte stage is represented by a long stretch along the UMAP (clusters 5, 78, 40, 41, 51). This is underscored by the long gap in detection of *CycB* mRNA representing these clusters in the UMAP (*Figure 2I and L*). However, in terms of physical space, in situ hybridization for *CycB* revealed only a relatively narrow gap near the boundary between spermatogonia and spermatocytes (*White-Cooper et al., 1998*). The UMAP territory for young spermatocytes may be stretched in gene expression space because this stage is a time of extensive and dynamic changes in gene expression, with many genes dramatically upregulated as the spermatocyte expression program initiates, and many genes downregulated from the spermatogonial phase (*Figure 2L*; see also *Shi et al., 2020*). Meanwhile, the young spermatocytes do not take as much physical space as the more mature stages, since they are smaller in size and fewer in number (2462 nuclei in clusters 5, 78, 40, 41, 51, compared to ~4,100 nuclei in clusters 35, 33, 48, 105, 64, 45, 13, 56,16).

## The spermatocyte transcription program

The spermatocyte period features onset of dramatic transcriptional changes. Many genes expressed in spermatocytes are transcribed in few or no other known cell types, including the markers *kmg*, *Rbp4*, *fzo*, *can*, *sa* (see references in *Table 1*). This robust onset of cell type-specific transcription appears as an increase in the number of different genes detected per nucleus (*Figure 3A and C*), leading to a substantial increase in transcriptome complexity. Coincident with this was a large increase in the number of unique molecular identifiers (UMI) scored, peaking in mid-to-late spermatocyte nuclei (clusters 35 and 33), with average UMI per cluster increasing from <5000 to>30,000 UMI per nucleus as spermatogonia differentiated to late spermatocytes (*Figure 3B and C*). These observed increases in the number of genes and UMIs detected were independently observed in a single-cell testis dataset to be introduced below (*Figure 3—figure supplement 1B and C*). The FCA paper noted that testis, heart, fat body, Malpighian tubules, and male reproductive glands had relatively high RNA levels and number of genes expressed compared to other tissues (*Li et al., 2022*). Reanalysis showed that mid-to-late spermatocyte nuclei exhibited the highest complexity of all, with average expressed gene (6000 compared to 2000) and UMI (30,000 compared to <20,000) numbers higher than for any cluster in the tissues noted by the FCA paper (*Figure 3—figure supplement 1A*). High transcriptome complexity has also been noted in mammalian spermatocytes (*Soumillon et al., 2013*).

After peaking in clusters 35 and 33, UMI values per nucleus decreased through clusters 48, 105, 56, 106, 21, where the 'tail' meets the 'torso' of the shark (*Figure 3B and C* - See *Figure 2A* for positions of numbered clusters), consistent with the observed lower expression of spermatocyte marker genes (*Figure 2L*). In the shark's upper torso and head, many clusters had very low UMI (*Figure 3C*), making developmental order difficult to assign. This is reflected in the UMAP shape, with clusters grouped rather than extended along a string as in early germ cell stages. We surmise these nuclei represent early spermatids, as classic studies showed that transcription falls dramatically from shortly before onset of the meiotic divisions, with no bulk incorporation of radioactive uridine detected in haploid round and early elongating spermatid nuclei (*Gould-Somero and Holland, 1974*; *Olivieri and Olivieri, 1965*). Although UMI counts and average number of genes expressed in the snRNA-seq were low in post-meiotic clusters (*Figure 3C*), nonetheless, post-meiotic transcription appeared more extensive than previously appreciated, with transcripts from approximately 1000 genes detected.

Spermatocytes showed sex chromosome specific trends in gene expression changes. Overall, Y-linked transcripts were strongly upregulated in spermatocytes (*Figure 3D*), primarily driven by the robust expression of 8 of the 12 single copy genes. For example, transcription of the Y-linked fertility factors *kl-3* and *kl-5*, which encode flagellar dyneins expressed only in male germ cells, was massively upregulated (125 and 275 fold respectively, with similarly large differences in absolute expression).

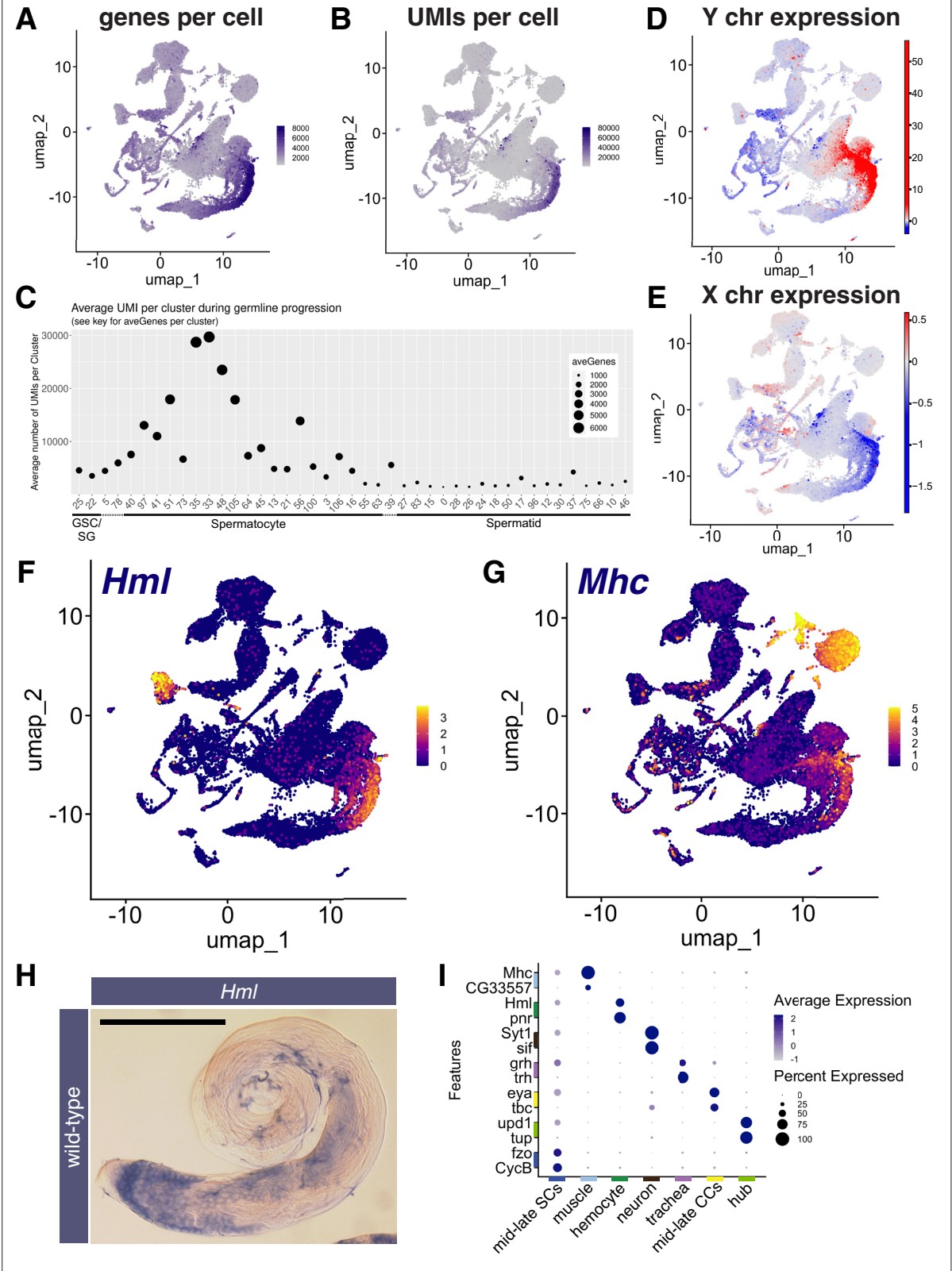

**Figure 3.** Features of the spermatocyte transcription program. (**A, B**) UMAPs of snRNA-seq data showing: (**A**) number of genes detected as expressed and (**B**) Number of unique molecular identifiers (UMIs) detected per nucleus. (**C**) Plot of average number of genes expressed (dot size) and UMIs detected per nucleus per germline-annotated Leiden 6.0 cluster, ordered by estimated progression of germ line differentiation. (**D, E**) UMAPs of snRNA-seq data showing average expression of: (**D**) Y chromosome tor (**E**) X chromosome encoded transcripts relative to an expression-matched control set

*Figure 3 continued on next page*

*Figure 3 continued*

(gene sets with binned expression matching transcript lists). Note the dramatically different relative expression scales in (**D**) vs (**E**) UMAPs taken directly from ASAP (*Li et al., 2022*; *Gardeux et al., 2017*). (**F, G**) UMAP plots of snRNA-seq raw counts (log-transformed) showing expression of: (**F**) *Hemolectin* (*Hml*) in hemocytes and late spermatocytes and (**G**) Myosin *heavy chain* (*Mhc*) in muscle and late spermatocytes. Yellow: relative expression high. (**H**) Testis hybridized in situ with biotinylated antisense RNA probe to *Hml*, showing expression (blue) in spermatocytes. Scale bar, 200 µm. (**I**) Dot Plot showing expression of pairs of tissue specific markers across cell types in the testis plus seminal vesicle snRNA-seq sample. Average expression of each gene in a given cell type (here, a cluster at Leiden 0.4 resolution) denoted by color intensity. Percent of nuclei of the given cell type scored as expressing each gene denoted by dot size. Colors along axes (see *Figure 1B*) indicate the relevant cell type for tissue-specific marker pairs. (see also *Figure 3— figure supplements 1 and 2* and *Figure 3—source data 1*).

The online version of this article includes the following source data and figure supplement(s) for figure 3:

**Source data 1.** Average expression of 496 transcription factors in Leiden 0.4 clusters from *Li et al., 2022*.

**Figure supplement 1.** Magnitude of the mid-to-late spermatocytes transcription program.

**Figure supplement 2.** Markers of other tissues also expressed in mid-to-late spermatocytes.

As 10 X sequencing utilizes oligo(dT) primers, the late appearance of reads from the Y-linked fertility factors in spermatocytes may reflect the very long time required to complete synthesis of the mature transcripts, which have extremely large introns (*Fingerhut et al., 2019*). For X-linked genes, analysis of the snRNA-seq data showed a similar level of expression relative to a control set of genes from all chromosomes in spermatogonia and early spermatocytes. However mid-to-late spermatocytes featured a transition to reduced expression of X-linked genes relative to the control set (*Figure 3E*), consistent with the roughly twofold lower expression of X linked genes compared to generally expressed autosomal genes observed previously (*Mahadevaraju et al., 2021*).

One surprise that emerged from the UMAP geography was that later stage spermatocytes split into three parallel streams, all expressing spermatocyte-specific markers. (*Figure 2A*). Strikingly, nuclei in the leftmost and middle streams (clusters 64 and 45, respectively) had considerably lower UMI count than in the robust mainstream (cluster 35; *Figure 3B and C*). The cause underlying such different UMI levels among late spermatocytes is not known, but could suggest a stochastic component to meiotic chromosome condensation and the attendant chromosome-wide downregulation in gene expression.

A second notable feature was the expression in mid-to-late spermatocytes of many markers classically associated with other cell types. Notably, markers for hemocytes (*Hemolectin – Hml*), muscle (*Myosin heavy chain - Mhc*), neurons (*Synaptotagmin - Syt1*), and epithelial cells (*grainy head - grh*) selected as identifiers of these cell types in the FCA study of adult *Drosophila* tissues (*Li et al., 2022*), were upregulated in late spermatocytes (*Figure 3F, G1*, *Figure 3—figure supplement 2A, B*). Similar upregulation of *Mhc*, *Hml*, *grh,* and *Syt1* in spermatocytes was independently observed in a single cell testis dataset introduced below (*Figure 3—figure supplement 2E*), so is not likely to be an artifact of isolation of nuclei. Other mRNAs normally thought of as markers of somatic cells were revealed to be upregulated in mid-to-late spermatocytes in the snRNA-seq dataset, such as *eyes absent* (*eya*) and *unpaired 1* (*upd*; *Figure 3—figure supplement 2C and D*).

Expression in spermatocytes was confirmed by in situ analysis for *Hml* in wild-type testes and under mutant conditions where spermatocytes accumulate (*Figure 3H*, *Figure 3—figure supplement 2F*). Additionally, the mRNAs for the late cyst cell markers *geko* and *Visual system homeobox 1* (*Vsx1*) and the cyst cell marker *eya* were directly detected in spermatocytes by FISH (*Figure 3—figure supplement 2G, H, I*; Figure 6K and S). Note that expression of the somatic marker mRNAs was generally lower in spermatocytes than in the marker somatic tissue (*Figure 3I*), as observed in *Vsx1* in situs, for example (*Figure 3—figure supplement 2H, I*). It will be fruitful to investigate the cause of the seemingly promiscuous expression in spermatocytes of certain somatic marker genes, whether their encoded proteins accumulate, and what role these genes may have in spermatocyte biology.

Although spermatocytes feature an overall increase in UMIs per cell, spermatocytes do not appear to simply be 'permissive' for general expression. For example, mRNA for the cyst marker *tj* was not detected in spermatocytes (*Figure 1D*). This selectivity argues against the possibility that observing 'somatic markers' in spermatocytes represents some technical artifact. In addition, some transcription factors that mark specific somatic tissues were detected as upregulated in spermatocytes compared to spermatogonia, while others were not., Across all adult fly tissues, the FCA project identified 496 transcription factors predicted to have a high tissue specificity score (*Li et al., 2022*). That analysis predicted 351 of these as expressed in certain somatic cell types but not in germ line (Table S3 from *Li*

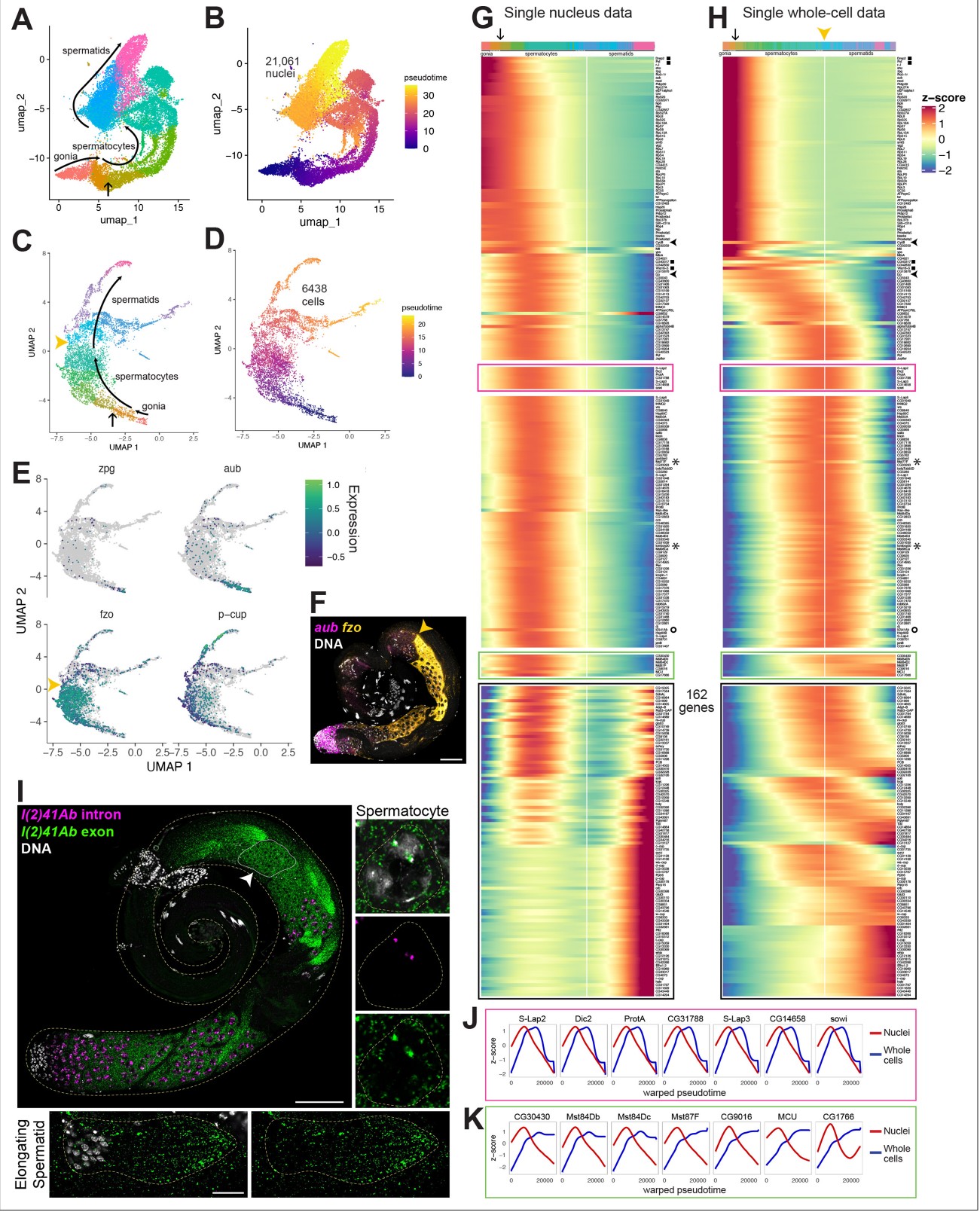

**Figure 4.** Developmental transitions revealed by comparing sn and scRNA-seq. (**A–D**) UMAP plots of germline-annotated data from (**A, B**) FCA snRNA-seq of adult testis plus seminal vesicle and (**C, D**) whole-cell scRNA-seq of adult testis. (**A, C**) Color denotes germline differentiation stage, with clustering in A as in *Figure 4—figure supplement 1*. (**B, D**) Color denotes pseudotime, with the few nuclei lacking a calculable pseudotime value colored gray. (**E**) UMAPs of scRNA-seq data showing log10(Expression) levels of cell-stage diagnostic markers *zpg*, *aub*, *fzo*, and *p-cup*. (**F**) FISH

*Figure 4 continued on next page*

*Figure 4 continued*

of diagnostic genes *aub* and *fzo*. Scale bar, 50 µm. Yellow arrowhead in (**C, F, H**) marks the point at which *fzo* transcript drops, annotated as early round spermatids based on (**F**). (**G–H**) Heatmaps of row-normalized (z-score) gene expression over pseudotime for: (**G**) all germline-annotated single nuclei from panel A, (**H**) all germline-annotated single whole cells from panel C, with genes in same order as in G. X axes, pseudotime; Y axes, genes. Vertical white line: nuclei (**G**) or cells (**H**) where level of *fzo* mRNA has dropped 50% from peak to nadir (0 on Z score). *Fzo* transcript drop validated in the cell data, and marked with yellow arrowhead. Top bars: cell identity for each column, colored as in panels A,C. Arrow points to transition between clusters dominated by expression of genes up in spermatogonia to clusters dominated by expression of genes upregulated in early spermatocytes (see *Figure 4—figure supplement 1A and B*). Gene markers as follows: black filled squares, transcripts validated in *Figure 2—figure supplement 1*. Arrowheads, *CycB* and *fzo*. Asterisks, *Mst77F* and *tomboy20*. Circle, *l(2)41Ab*, validated in panel I. Black boxes: genes transcribed post-meiotically (see *Figure 5*). (**I**) FISH of *l(2)41Ab* intron (magenta) and *l(2)41Ab* exon (green). Arrowhead on whole testis points to a round spermatid cyst, which lacks intronic signal. High magnification insets: Side - Spermatocyte nucleus showing intronic and exonic probe signal. Bottom - elongating spermatid cyst showing exonic probe signal only. Scalebars: whole testis, 50 µm; spermatocyte, 5 µm; spermatid, 20 µm. (**J-K**) Comparison of gene expression over warped pseudotime for: (**J**) genes outlined by pink boxes in G,H; (**K**) genes outlined by green boxes in G,H. (see also *Figure 4—figure supplement 1*).

The online version of this article includes the following figure supplement(s) for figure 4:

**Figure supplement 1.** Developmental transitions revealed by trajectory analysis.

*et al., 2022*). Our analysis of the testis plus seminal vesicle portion of the FCA data showed that about a fifth (76) of these 'somatic' factors were upregulated in spermatocytes compared to spermatogonia, but most (275) exhibited no upregulation (*Figure 3—source data 1*).

## sn vs. scRNA-seq: dynamics of active transcription vs. stored RNAs

One of the regulatory mechanisms used to control cell differentiation in tissues is the programmed storage of mRNAs to be used later, long after the genes encoding these mRNAs are silenced. Spermatogenesis in *Drosophila* is especially conducive to the study of this phenomena because early spermatids, although largely transcriptionally silent, carry numerous cytoplasmic transcripts, many of which are recruited to be translated for temporal control of protein expression during spermatid morphogenesis (*Schäfer et al., 1990*; *Schäfer et al., 1995*). To identify transcripts under such control, we took advantage of the fact that single nucleus RNA sequencing detects recently transcribed or nuclear resident transcripts, whereas single cell RNA sequencing (scRNA-seq) samples a largely different RNA population. Therefore, we generated single cell RNA-seq of adult testes (without seminal vesicle), producing data for 6438 germ cells after quality control steps (Materials and methods).

The UMAP geography for both the snRNA-seq and scRNA-seq datasets showed progression from spermatogonia to spermatids, with germline differentiation classes present in sequential order (*Figure 4A–D, Figure 4—figure supplement 1*). In the scRNA-seq UMAP, as for snRNA-seq, expression of *zpg* marked a small number of spermatogonia located at the bottom tip, *aub* marked those same plus additional cells, presumably early spermatocytes, *fzo* marked largely differentiating spermatocytes with expression abruptly ending in early spermatids (yellow arrowheads in *Figure 4C and E*), and *presidents-cup* (*p-cup*) marked later elongating spermatids in an arm extending from the top of the UMAP (*Figure 4E*). Corroborating these expression patterns, FISH to whole mount testes clearly showed *aub* expression in spermatogonia and early spermatocytes, and *fzo* expression beginning in spermatocytes (*Figures 2B and 4F*). Notably, *fzo* mRNA was abruptly downregulated in early round spermatids soon after the second meiotic division (*Figure 4F*, yellow arrowhead).

Trajectory inference can assign a differentiation distance parameter to cells inferred from transcriptional differences, with distance noted as 'pseudotime' (*Trapnell et al., 2014*). Applying trajectory inference independently to the snRNA-seq and scRNA-seq germ line datasets produced contiguous trajectories. Using Monocle3, 99.9% (21,061/21,091) of the snRNA-seq germline nuclei were connected (*Figure 4B*). Notably, unlike prior trajectory analysis using Slingshot (*Li et al., 2022*), the inferred trajectory was contiguous, connecting cells of all differentiation points from early spermatogonia to late spermatids. Likewise, Monocle3 analysis of the 6438 germline cells from the scRNA-seq also produced a contiguous trajectory from spermatogonia to elongating spermatids (*Figure 4D*), although it did include a late bifurcation, the explanation of which may be technical or biological. For both datasets, pseudotime staging paralleled the ordered trajectory deduced from marker gene expression in UMAP clusters (*Figure 2*, *Figure 4A–E*).

Plotting normalized gene expression across pseudotime in the sn- and scRNA-seq datasets revealed both shared and contrasting dynamics. In both datasets, the same set of genes were expressed in

early germ cells, with expression diminishing over pseudotime (*Figure 4G and H*; red/orange in the top color bar, ending at the point indicated by an arrow, corresponding to the colors and arrow in *Figure 4A and C*). Similarly, some genes, including *fzo* and *CycB* (*Figure 4G and H* - black arrowheads on side) reached peak expression in spermatocytes and dropped to low levels by early spermatid stages in the scRNA-seq (*Figure 4E and H*, white vertical line marking *fzo* drop to z-score 0), consistent with in situ hybridization data (*Figure 4F* and published in *White-Cooper et al., 1998*, respectively).

Interestingly, in the snRNA-seq dataset, a group of over 200 genes in the middle region of the heat map reached peak expression in the mid-spermatocyte stages (green-aqua hues in the top color bar) then dropped in expression, falling halfway to their nadir at a point similar to the drop in *fzo* and *cycB* expression (vertical white line) (*Figure 4G*). In the scRNA-seq dataset, however, these same genes were still at or near peak mRNA accumulation in the same cells in which expression of *fzo* and *cycB* had already dropped (*Figure 4H*, white line). Thus, it is the *comparison* of the datasets that is most revealing: these genes are transcribed in spermatocytes, then transcription halts (inferred from the snRNA-seq dataset), but their mRNAs remain high in spermatids (inferred from the scRNA-seq dataset) well past the stage when *fzo* and *cycB* mRNAs disappear (inferred from both datasets). Several genes from this set, including *Male-specific transcript 77* F (*Mst77F*) and *tomboy20* (asterisks), have been previously demonstrated by in situ hybridization to maintain abundant transcripts in both spermatocytes and elongating spermatids (*Barckmann et al., 2013*; *Hwa et al., 2004*). Comparison of the snRNA-seq and scRNA-seq datasets suggests that such a pattern, previously described for a small number of transcripts, is shared by hundreds of genes.

We chose one such novel predicted perdurant transcript, *l(2)41Ab* (*Figure 4G–H*, marked with a circle), proposed to be involved in axoneme assembly in spermatids (*Zur Lage et al., 2019*), to validate in vivo. Signal from RNA FISH probes designed against a *l(2)41Ab* intron was clearly present in nuclear foci in spermatocytes, but absent from spermatid nuclei, suggesting that active *l(2)41Ab* transcription occurs in meiotic prophase (G2) and has ended by the early round spermatid stage (*Figure 4I*). Signal from probes designed against a *l(2)41Ab* exon, by contrast, was visible in the cytoplasm of spermatocytes and even more strongly present in early spermatids and the tails of elongating spermatids (*Figure 4I*). Such a pattern suggests that comparison of snRNA-seq and scRNA-seq datasets can correctly uncover novel genes that encode transcripts that remain in the cytoplasm for days after transcription has ceased.

Comparison of single cell and single nucleus data revealed several distinct classes of transcript behaviors in spermatids, each worthy of targeted follow-up study. Notably, perduring transcripts from the class of genes described above showed two types of behavior. For over a hundred genes, transcripts disappeared sharply in later elongating spermatids (7 genes with this pattern highlighted in pink box in *Figure 4G and H*). To plot transcript levels in the sn- and scRNA-seq datasets on the same X axes, a common 'warped' time scale was derived for the datasets (see Materials and methods). First, for all genes upregulated in spermatocytes, onset of transcription in the nuclear transcriptome was followed (with delay) by upregulation in the whole-cell transcriptome, generally dominated by cytoplasmic transcripts (*Figure 4I and J*). As seen in the heat map (*Figure 4H*), the graphs for the pink box genes show that the mRNAs remained at peak levels considerably later in the scRNA-seq than in the snRNA-seq data, but the transcripts were eventually strongly downregulated by late spermatid stages (*Figure 4J*: blue lines). This suggests complexity in mRNA regulation in the cytoplasm: stable maintenance in early spermatids and abrupt degradation in later spermatids, perhaps once transcripts have been translated. Interestingly, the protein products of several of these genes are present and functional in late spermatids and sperm (*Jayaramaiah Raja and Renkawitz-Pohl, 2006*), suggesting these proteins are actively maintained in the absence of new translation.

A second type of behavior was noted for a much smaller group of genes (green box in *Figure 4G and H*, graphed in K) where transcripts perdured even longer, remaining high through the latest stages assessed by scRNA-seq (blue lines in *Figure 4K*). As differentiation of late spermatocytes to late spermatids takes days (*Chandley and Bateman, 1962*), these remarkable transcripts maintained high levels of cytoplasmic abundance, with almost no sign of degradation, even days after active transcription had dropped off. This suggests exceptional stability, likely provided by specialized RNA-binding proteins. Some such transcripts, encoded by *Mst84Db*, *Mst84Dc*, and *Mst87F*, have long been recognized to be translationally regulated, with perdurance in the cytoplasm for up to 3 days (*Kuhn*

*et al., 1991*; *White-Cooper et al., 1998*). Others, including *CG30430, CG9016, MCU*, and *CG17666* have not been previously reported to undergo translational regulation. The differences in degradation timing revealed by scRNA-seq (*Figure 4G–H*, pink vs green boxes) may hint that distinct groups of RNAs, and thus their protein products, are engaged at different stages of spermatid morphogenesis.

Another compelling example of the utility of comparing snRNA-seq and scRNA-seq data is highlighted by the group of genes outlined in black (*Figure 4G and H*). In the snRNA-seq dataset, these genes are expressed in spermatocytes but transcription shuts down in early spermatids and remains off for a considerable period before expression is activated again in mid-to late elongation stages. Thus, few transcripts bridge the gap between late spermatocytes and mid-stage elongating spermatids, as if the two stages were disconnected. In contrast, in the scRNA-seq dataset, many of these same genes showed continued high transcript levels throughout the spermatid stages, presumably representing storage of mRNAs in the cytoplasm. In consequence, the mature spermatocyte to elongating spermatid stage transcriptomes were well connected through a smooth gradient of transcript levels in scRNA-seq data (*Figure 4H*, black box).

## Reactivation of transcription in mid-to-late elongating spermatids

The ability of snRNA-seq to highlight dynamic transcriptional changes during cellular differentiation revealed striking transcriptional (re)activation of a subset of 162 genes in mid-to-late elongating spermatids, a phenomenon previously described for only 24 genes, called 'post-meiotic transcripts' (*Barreau et al., 2008*). Analysis for genes specifically enriched in late pseudotime identified a list of 162, here termed spermatid transcribed genes (*Figure 4G*, black box, and *Figure 5—source data 1*). These included 18 of the previously identified 24. FISH revealed *flyers-cup* (*f-cup*) RNA at the distal end of elongated spermatid bundles, as expected (*Barreau et al., 2008*, *Figure 5A and B*; *Figure 5—figure supplement 1A*). RNA from *loquacious* (*loqs*), a newly identified spermatid transcribed gene, was similarly localized (*Figure 5C*; *Figure 5—figure supplement 1B*). Transcripts from *walker cup* (*wa-cup*) and *scotti* (*soti*) also localized to the distal ends of elongating spermatids as expected (*Barreau et al., 2008*, *Figure 5D–F*; *Figure 5—figure supplement 1C, D*). Analysis of earlier elongating spermatid cysts by single molecule RNA FISH (smFISH) supported active transcription of *wa-cup* and *soti* in spermatids: smFISH revealed foci in spermatid nuclei, suggesting nascent, post-meiotic transcription (*Figure 5E and F*, arrows), as well as perinuclear granules (*Figure 5E and F*, arrowheads), which could represent newly synthesized RNAs being trafficked toward the distal ends of the spermatids.

Analysis of the snRNA-seq data showed that many newly identified spermatid transcribed genes, including *Pp2C1* and *CG6701*, were initially expressed in spermatocytes or spermatogonia, downregulated in early spermatids, and later reactivated during mid-to-late elongation (*Figure 4G*-upper half of black box; *Figure 5G and H*). Other newly identified spermatid transcribed genes, including *Parp16* and *Glut3,* were weakly expressed in spermatocytes but robustly transcribed in elongating spermatids (*Figure 4F* - lower half of black box; *Figure 5G and H*). Both patterns are consistent with RT-qPCR and RNA in situ hybridisation results for the 24 post-meiotic transcripts previously identified (*Barreau et al., 2008*). Together the results show two sources of RNAs in elongating spermatids: cytoplasmic perdurance of RNAs transcribed in spermatocytes (*Figure 4H*), and de novo post-meiotic (re) activation of transcription of certain genes (*Figure 4G*, *Figure 5G and H*).

The majority of the spermatid transcribed genes remain functionally uncharacterized, and await investigation. GO term analysis showed no significant enrichment for any single biological process or pathway, although several functional classes were represented (*Figure 5—source data 1*). Additionally, genes in this set did not appear to be coordinately reactivated, as by a single regulatory circuit. Rather, the likelihood of genes to be (re)activated concordantly was weakly correlated with their expression level, with a few outliers (*Figure 5—figure supplement 1E*).

## Progression of differentiation in the somatic cyst cell lineage

Somatic cyst cells govern many germline transitions, from stem cell behavior through sperm maturation and release (*Figure 1A*; *Figure 6O*). The snRNA-seq approach may be especially useful for characterizing cyst cell transcriptomes across differentiation stages because the long, thin, extended shape of many cyst cell types may make isolation of intact cells difficult.

Cyst lineage identity was assigned by expression of three transcriptional regulators, *traffic jam* (*tj*), *eya* and *sine oculis* (*so*) (*Figure 1B and D*; *Figure 6A, B and U*; *Figure 6—figure supplement 1C and*

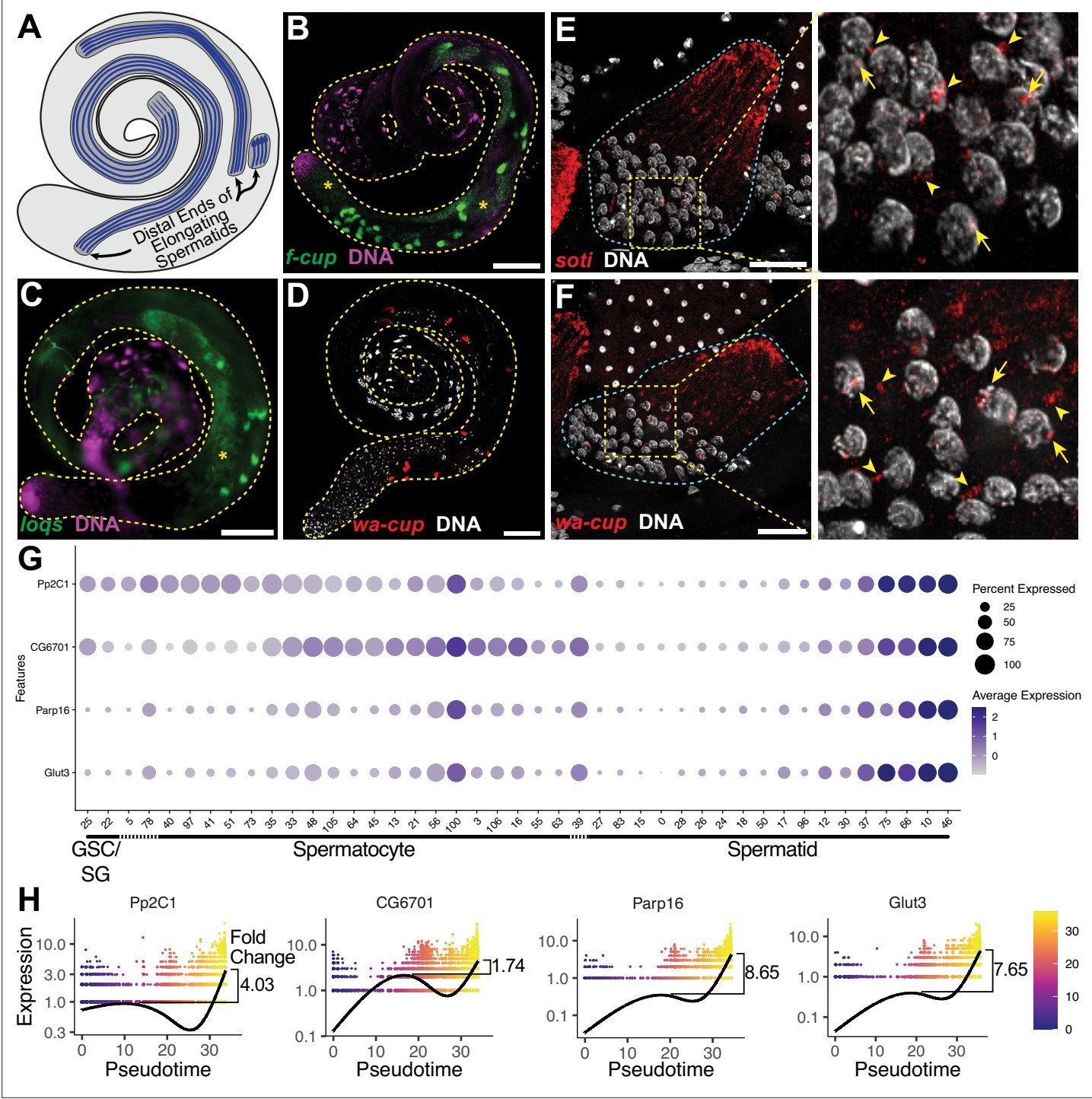

**Figure 5.** The transcript landscape of haploid spermatids. (**A**) Diagram of spermatid orientation in the testis at different stages of spermatid elongation. Arrows: distal ends of spermatid cysts. (**B, C**) RNA FISH of representative transcripts (**B**): *f-cup*, (**C**): *loqs* in whole testes showing different patterns of mRNA localization for post-meiotically transcribed genes. DNA (magenta), target RNA (green). Asterisk: signal in spermatocytes as surmised by relatively large nuclear profile revealed by absence of signal. Bar: 100µm. (**D**) smFISH for *wa-cup* (red) in whole testes. DNA (white). Bar: 100µm. (**E, F**) Left: smFISH for *soti* (**E**) or wa-cup (**F**) in a single early elongating spermatid cyst (cyan dashed outline). RNA (red) and DNA (white). Bar: 25 µm. Right: Enlarged image of yellow dashed box showing spermatid nuclei. Arrows: nuclear transcripts. Arrowheads: perinuclear granules. (**G**) Dot plot for selected spermatid transcribed genes showing expression levels in each germ cell cluster. (**H**) Expression over pseudotime for selected spermatid transcribed genes and fold change between late pseudotime and previous expression maxima in early/mid pseudotime (see also ***Figure 5—figure supplement 1***).

The online version of this article includes the following source data and figure supplement(s) for figure 5:

*Figure 5 continued on next page*

*Figure 5 continued*

**Source data 1.** List of 162 spermatid transcribed genes.

**Figure supplement 1.** Reactivation of transcription in spermatids.

*D*). While both very early clusters (62, 36, 58) and late clusters (79, 68 and 80, 84, 72) were simple to describe (see below), the middle region of the 2-D UMAP presented a tangle. However, re-projecting the lineage to preserve a third dimension clarified the assignment of cluster order through the middle region (*Figure 6C*; see Materials and Methods; rotatable 3D representation available at https://doi.org/10.5061/dryad.m63xsj454). This, combined with pseudotemporal ordering (*Figure 6—figure supplement 1A and B*) and marker analysis, enabled assignment of specific clusters to the cyst cells associated with stem cell, spermatogonial, spermatocyte, spermatid or sperm release stages of germ-line development. Perhaps notably, the few clusters difficult to assign had relatively low UMI count (*Figure 6E*; clusters 81, 94, 95, 67, 60).

The cyst lineage begins with proliferative *stg*-expressing CySCs (Cluster 62, *Figure 1E*; *Figure 6A and D*). Expression of *tj* suggested that clusters 36 and 58 represent early-stage cyst cells that enclose transit amplifying spermatogonia (*Figure 6B and U*) labeled SgCC in *Figure 6D*, as Tj protein marks the nucleus of these early cyst cells but was not detected by immunofluorescence staining in spermatocyte-associated cyst cells (ScCC) (*Li et al., 2003*; *Zoller and Schulz, 2012*). Complementing this, *CG3902* mRNA was also detected in the same clusters as *tj*, and a CG3902 protein trap line revealed cytoplasmic protein accumulation up to early ScCC and no detectable protein thereafter (*Figure 6U*; *Figure 6—figure supplement 1E and F*). *P-element induced wimpy testis* (*piwi*) mRNA was enriched in the same clusters as *tj and CG3902*, and also was detected at lower levels in nuclei of subsequent clusters (for example, clusters 47 and 77; *Figure 6D* SgCC and ScCC, F,U). FISH and analysis of a Piwi protein trap confirmed expression in early cyst cells associated with spermatogonia at the testis apex, as well as in cyst cells enclosing growing spermatocyte cysts (*Figure 6G–I*, arrowhead and dashed arrow, respectively). Interestingly, *piwi* transcripts were also detected in more mature cyst cells associated with elongated and polarized spermatids (*Figure 6G*, solid arrow), highlighting the differences between active transcription detected by snRNA-seq and perdurance of cytoplasmic RNA detected by FISH.

Both *tj* and *eya* mRNA expression was detected in cluster 58, but onward, *tj* mRNA was abruptly down regulated while *eya* transcript expression increased. We surmise this marks the transition to cyst cells associated with spermatocytes, labeled ScCC in *Figure 6A, D, J, U*, since Eya protein is known to accumulate in cyst cell nuclei from late stage spermatogonia to spermatocytes (*Fabrizio et al., 2003*). FISH confirmed the cyst cell *eya* pattern (*Figure 6K*, arrowhead and dashed arrow), while also revealing accumulation of *eya* transcript in cyst cells associated with post-meiotic spermatids (*Figure 6L*, solid arrow). Additionally, sparser but detectable *eya* transcript was found in spermatocytes themselves, corroborating our observation that it is one of several somatic markers expressed in spermatocytes (*Figure 3I*; *Figure 6K*, white outline).

Intriguingly, the cyst lineage bifurcates after cluster 58, with clusters 77, 65, and 98 successively in one arm and 47, 104, and 88 in the other (*Figure 6A, C and D*). This might be due to onset of differentiation of head versus tail cyst cells, and would represent the first hint at when this occurs in the lineage. Identification and characterization of genes differentially expressed within the split could reveal whether cyst cells specific to a given arm of the lineage govern different properties of spermatocyte-containing cysts.

Expression of *Amphiphysin* (*Amph*) supported the conclusion that the bifurcation in the cyst cell lineage after cluster 58 represents cyst cells associated with spermatocytes. *Amph* mRNA was high before the bifurcation and persisted at lower levels in the two arms of the split, dropping substantially by cluster 74 (*Figure 6M and U*). Expression of a protein trap confirmed that *Amph* protein levels were high in SgCCs and ScCCs (*Figure 6N*, arrowhead and dashed arrow, respectively) and declined significantly in cysts containing early spermatids (data not shown). It is intriguing that the snRNA-seq and protein trap indicate that Amph expression is strongest in early cyst cells, even though it encodes a BAR domain protein required to form the actomyosin clamp that maintains head cyst cell membrane integrity as these cells wrap around spermatid heads late in spermatogenesis (*Kapoor et al., 2021*).

Branches of the cyst lineage rejoin at cluster 74 implying a transition to a common transcriptional state. Interestingly, comparison of *Akr1B* vs. *Amph* suggests that cluster 74 contains cyst cells

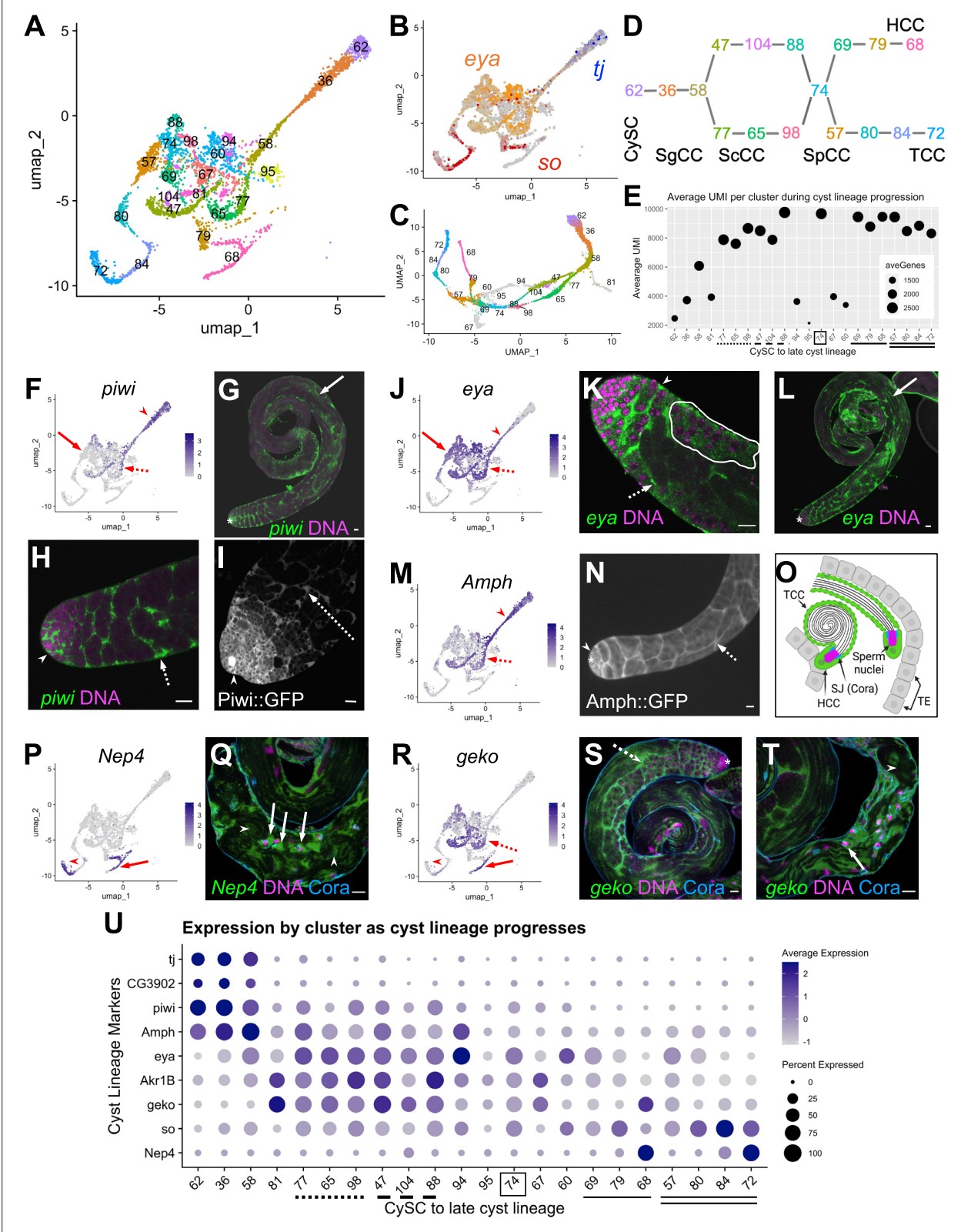

**Figure 6.** Differentiation in the somatic cyst cell lineage. (**A**) Cyst cell lineage portion of the UMAP from snRNA-seq data with Leiden 6.0 cluster numbering. (**B**) Expression of *tj* (blue), *eya* (orange), and *so* (red) projected on the UMAP (heatmaps in *Figure 6—figure supplement 1C and D*). (**C**) 2D UMAP of cyst cell clusters newly reprojected in their own gene expression space (Materials and methods; note different axis coordinates relative to panel A). Cluster colors correspond to panel A except for unidentifiable clusters with low UMIs (gray). (**D**) Schematic of cyst cell cluster progression

*Figure 6 continued on next page*

*Figure 6 continued*

from CySC, to spermatogonia-associated (SgCC) to spermatocyte-associated (ScCC) to spermatid-associated (SpCC) inferred from the 3D UMAP in panel C. Cluster numbers and colors as in A. Note two splits, one earlier and one later in cyst lineage progression. (**E**) Plot of average number of genes (dot size) and average number of UMIs per cyst lineage-annotated cluster; with clusters ordered by deduced progression of differentiation. Dotted and dashed lines under cluster numbers represent the early split shown in D, while single (HCC) and double (TCC) solid lines represent the later split. (**F - T**) UMAPs compared to FISH (RNA), IF or live GFP emission (from tagged protein) images. For each gene comparison, arrows, dashed arrows and/or arrowheads point out the same cell type in the UMAP and its corresponding FISH and/ or GFP fluorescence image. (**F**) *piwi* mRNA expression projected on the UMAP. (**G, H**) *piwi* mRNA (FISH; green) and DNA (magenta). (**G**) whole testis, (**H**) testis apical tip. (**I**) Apical tip of testis expressing GFP tagged Piwi protein. (**J**) *eya* mRNA expression projected on the UMAP. (**K, L**) *eya* mRNA (FISH; green) and DNA (magenta). (**K**) Testis apical tip. (**L**) Whole testis. Outline demarcates cyst of spermatocytes showing *eya* transcript expression (see *Figure 3* and its supplement). (**M**) *Amph* expression projected on the UMAP. (**N**) Apical third of a testis expressing GFP-tagged Amph protein. (**O**) Schematic of head cyst cells (HCC, green solid outline) embedded (left) or not embedded (right) in the terminal epithelium (TE, gray), with the tail cyst cell (TCC, green dashed outline) extending away, containing either a pre-coiled (right) or coiled (left) spermatid bundle (Sperm nuclei, magenta). Coracle (Cora, blue) marks the septate junction (SJ) between TCC and HCC. (**P**) *Nep4* mRNA expression projected on the UMAP. (**Q**) Testis base showing *Nep4* mRNA (FISH, green), DNA (magenta), and Cora (septate junctions, cyan). Arrowhead points to TCCs associated with coiled sperm. (**R**) UMAP of *geko* mRNA expression. (**S, T**) *geko* mRNA (FISH, green), DNA (magenta), and Cora (septate junctions, cyan). (**S**) Whole testis. (**T**) Testis base. Arrowhead points to TCCs associated with coiled sperm. (**U**) Dotplot of gene expression (Y axis) by cluster as cyst lineage progresses through differentiation (X axis, left to right). Averaged expression in each cluster indicated by color scale. Percent of cells within a cluster expressing the gene indicated by size of dot. Lines under cluster numbers as in E. Asterisk denotes hub. Bars: 20 μm (see also *Figure 6—figure supplement 1*).

The online version of this article includes the following figure supplement(s) for figure 6:

**Figure supplement 1.** Cyst Lineage pseudotime and re-projection.

associated with early or elongating spermatids. Akr1B transcripts were elevated in both arms of the split through to cluster 74 (*Figure 6—figure supplement 1G*) while *Amph* expression had decreased by cluster 74 (*Figure 6M and U*). Analysis of a *Akr1B::GFP* protein trap confirmed its expression in cyst cells associated with elongating spermatid cysts (SpCC) (*Figure 6—figure supplement 1H1*, arrows). Thus, it appears that, with respect to the transcriptome, a developmental transition occurs within the cyst lineage as these cells mature from support of spermatocytes to early spermatid cysts.

After cluster 74, the lineage again splits, with marker analysis suggesting that this correlates with differentiation of late stage head cyst cells (HCC, clusters 69, 79, 68) versus putative tail cyst cells (TCC, clusters 57, 80, 84, 72; *Figure 6A, C and D*). Enrichment of *eya* was dramatically reduced in the clusters along either late cyst cell branch compared to those of earlier lineage nuclei (*Figure 6J and U*). However, *Neprilysin-4* (*Nep4*), a metalloprotease involved in male fertility (*Sitnik et al., 2014*), was upregulated in both late branches (*Figure 6P and U*). FISH-IF confirmed *Nep4* mRNA expression in late HCCs and TCCs associated with fully coiled spermatids (*Figure 6Q*, arrow and arrowheads, respectively) with Coracle staining defining the septate junctions between HCCs and TCCs. Intriguingly, the snRNA-seq data showed that *geko*, an olfactory gene not studied in the testis (*Shiraiwa et al., 2000*), was upregulated both in ScCCs and in part of cluster 68 (*Figure 6A, R and U*). In fact, higher resolution (Leiden 8.0) analysis divided cluster 68 into nuclei either enriched or not for *geko* (data not shown). FISH for *geko* revealed expression in the testis basal region with high expression in HCCs (*Figure 6T*, arrow) and lower expression in TCCs (*Figure 6T*, arrowhead) as predicted by the UMAP (*Figure 6R and U*). FISH signal was also observed in cyst cells associated with spermatocytes and elongating spermatid cysts throughout the testis (*Figure 6S*, dashed arrow). While TCC-specific transcripts remain elusive, exploring the putative HCC and TCC clusters further will not only test our tentative assignments, but also possibly reveal unique roles and processes carried out by these cells during spermatid retraction and coiling (*Figure 6O*).

## The hub: architectural organizer and key signaling center

The hub is a small group of somatic, epithelial-like cells at the testis apex that acts as a niche, providing signals that maintain GSC and CySC fate (*Hardy et al., 1979*; *Kawase et al., 2004*; *Tulina and Matunis, 2001*; *Kiger et al., 2001*; *Leatherman and Dinardo, 2010*; *Shivdasani and Ingham, 2003*). Initial marker analysis suggested that the hub maps to cluster 90. However, only 79 of these 120 nuclei were near to each other in 2D UMAP space, while other members of cluster 90 were dispersed (*Figure 7—figure supplement 1A*). The 41 outcast nuclei either expressed germline genes such as *Rbp4, zpg, p53,* or *vas* (*Figure 7—figure supplement 1B–E*) or rarely or only inconsistently expressed signature genes known to be enriched in hub cells (*Figure 7A–D*, dashed red circles). This

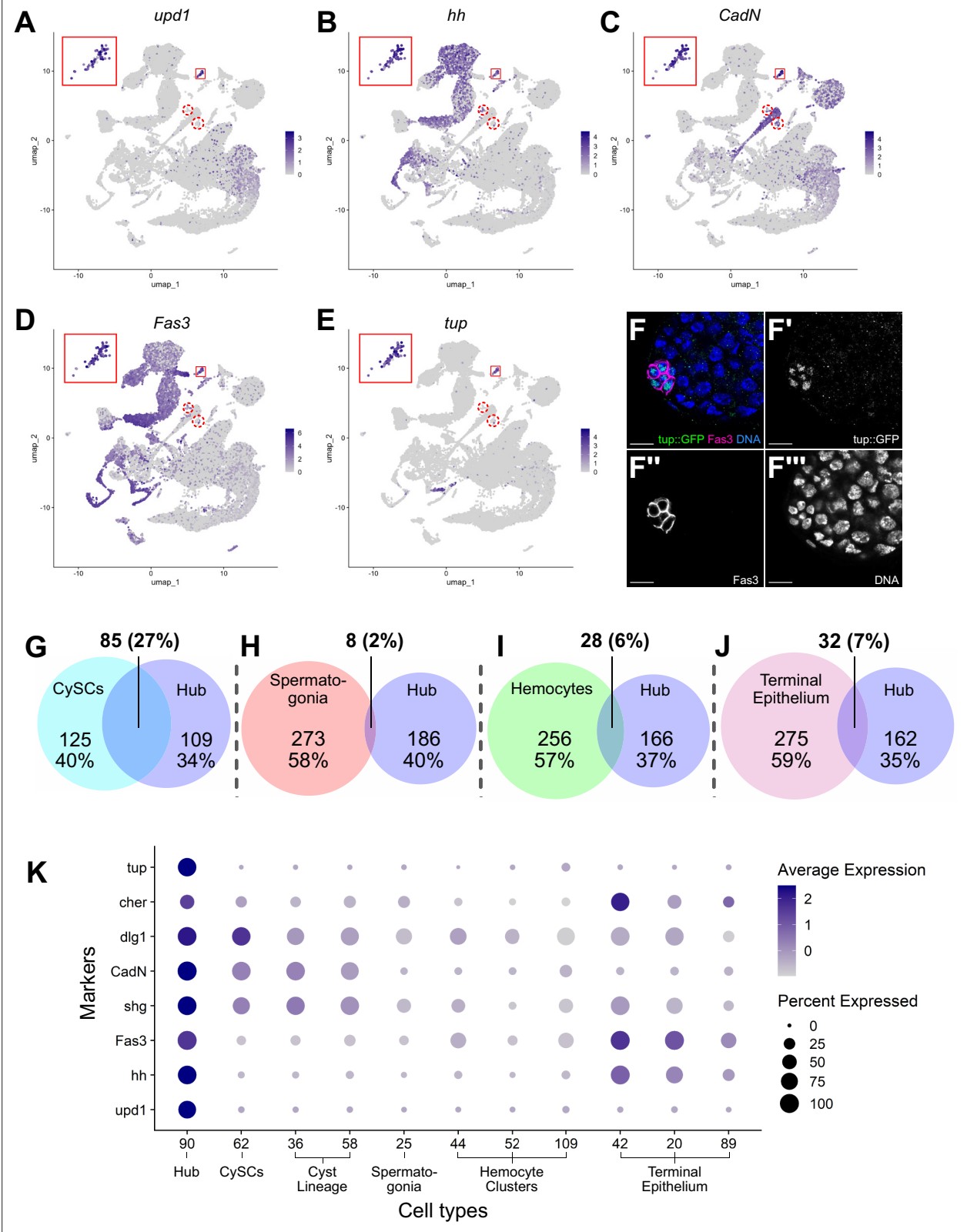

**Figure 7.** Characteristics of the Hub. (**A–E**) Expression of the indicated genes (*upd1*, *hh*, *CadN*, *Fas3*, *tup*) projected onto the testis snRNA-seq UMAP with the 79 definitive hub nuclei outlined (small red box), with reprojection of hub nuclei (larger red box). Color intensity corresponds to expression level, shown as normalized average logFC. Red dashed circles contain non-hub nuclei of cluster 90 (see text and **Figure 5**). (**F-F'''**) Apical tip of adult testis carrying a Tup::GFP fusion transgene revealing protein expression largely restricted to hub nuclei delimited by Fas3 (magenta). Nuclei (blue). Scale

*Figure 7 continued on next page*

*Figure 7 continued*

bar: 10 μm. (**G–J**) Paired Venn diagrams, comparing up-regulated genes in the hub vs. clusters containing either CySCs, spermatogonia, hemocytes, or terminal epithelial cells. Overlap in upregulated genes was greatest between Hub and CySCs. In each pairing, circle size reflects the number of genes compared for each cluster. Genes are listed in *Figure 7—source data 1*; the specific genes for hub vs. CySCs in *Figure 7—source data 2*. (**K**) Dot plot of expression of selected genes comparing hub to CySCs, the early Cyst lineage, Spermatogonia, Hemocytes, and Terminal epithelium (see also *Figure 7—figure supplements 1 and 2*).

The online version of this article includes the following source data and figure supplement(s) for figure 7:

**Source data 1.** Differentially expressed genes in Hub, CySCs, spermatogonia, terminal epithelium, or hemocytes.

**Source data 2.** Differentially expressed gene comparison between Hub and CySC.

**Figure supplement 1.** Hub justification using markers.

**Figure supplement 2.** Further support for Hub identity using alternate clustering methodology.

strongly suggests that they differ transcriptionally from the 79 tightly clustered nuclei. This conclusion was further supported by additional subclustering of the 120 nuclei by two independent methods (*Figure 7—figure supplement 2A–B*). Consequently, cluster 90 was pared down to 79 definitive hub nuclei.

The snRNA-seq data show that hub nuclei express genes involved in signaling as well as markers common in epithelial cells. For example, *upd1* and *hedgehog* (*hh*) are upregulated, consistent with the hub's role in stem cell maintenance (*Figure 7A, B and K*; *Tulina and Matunis, 2001*; *Kiger et al., 2001*; *Michel et al., 2012*; *Amoyel et al., 2013*). Additionally, the definitive hub cell nuclei showed enriched expression of mRNAs encoding proteins implicated in cell-cell adhesion/junctional complexes, including Fasciclin III (Fas3), E-Cadherin (Shg), N-Cadherin (CadN), Discs large (Dlg1), and Cheerio (Cher; an orthologue of Filamin), all of which proteins have been shown to mark hub cells (*Brower et al., 1981*; *Le Bras and Van Doren, 2006*; *Boyle et al., 2007*; *Tanentzapf et al., 2007*; *Papagiannouli and Mechler, 2009*; *Figure 7C, D, F and K*).

The definitive identification of hub nuclei allowed analysis for upregulated genes (log$_2$FC ≥1, compared to the full testis plus seminal vesicle dataset; *Figure 7—source data 1*) as new candidate hub markers. One such encodes the transcription factor Tailup (Tup, also known as Islet *Thor and Thomas, 1997*; *She et al., 2021*; *Boukhatmi et al., 2012*; *Figure 7E*). Indeed, a Tup::GFP transgene showed strong protein expression marking hub nuclei at the apex of adult testes (*Figure 7F–F'''*). This is consistent with recent evidence that Tup is expressed in and required for niche organization in the male embryonic gonad (*Anllo et al., 2019*; *Anllo and DiNardo, 2022*). The expression of Tup in the 79 'true hub' nuclei of the snRNA-seq dataset but not the other nuclei of the original 120 further validates the exclusion of the 41 dispersed nuclei previously included in cluster 90.

Previous lineage-tracing showed that hub cells and CySCs derive from a common pool of gonadal cells during embryogenesis (*DeFalco et al., 2008*; *Le Bras and Van Doren, 2006*; *Dinardo et al., 2011*). Comparison of genes up-regulated in hub cells with those up-regulated in CySCs strongly reflects this developmental relatedness. Hub and CySC (cluster 62) nuclei shared 27% of their up-regulated genes, likely reflecting their embryonic co-origin (*Figure 7G*, *Figure 7—source data 2*). In contrast, the fraction of shared upregulated genes was much lower between the hub and several lineally and functionally distinct cell types, including spermatogonia, hemocytes and terminal epithelia (*Figure 7H–J*). Supporting this, several genes up-regulated in hub nuclei were also highly expressed in the CySCs and the cyst lineage, but much lower in spermatogonia and hemocytes (*Figure 7K*). The transcriptional similarity observed in the adult cell types could account for the ability of hub cells to replenish CySCs after drastic injury and explain the shift of one lineage toward the other when specific gene functions are compromised (*Greenspan et al., 2022*; *Herrera et al., 2021*; *Hétié et al., 2014*; *Dinardo et al., 2011*; *Voog et al., 2014*).

## Epithelial cells of the testis organ

A key role of the terminal epithelial cells (TE) is to anchor the head end of elongated spermatid bundles at the base of the testis during individualization and coiling (*Figure 6O*) so that the sperm are positioned for release into the seminal vesicle (SV). Marker gene analysis suggested that both the TE and SV reside in the 'mermaid' of the UMAP (*Figures 1B and 8A*). snRNA-seq showed enrichment for *hh* in two broad areas of the UMAP 'mermaid' (*Figure 8B*; *Figure 8—figure supplement 1A*),

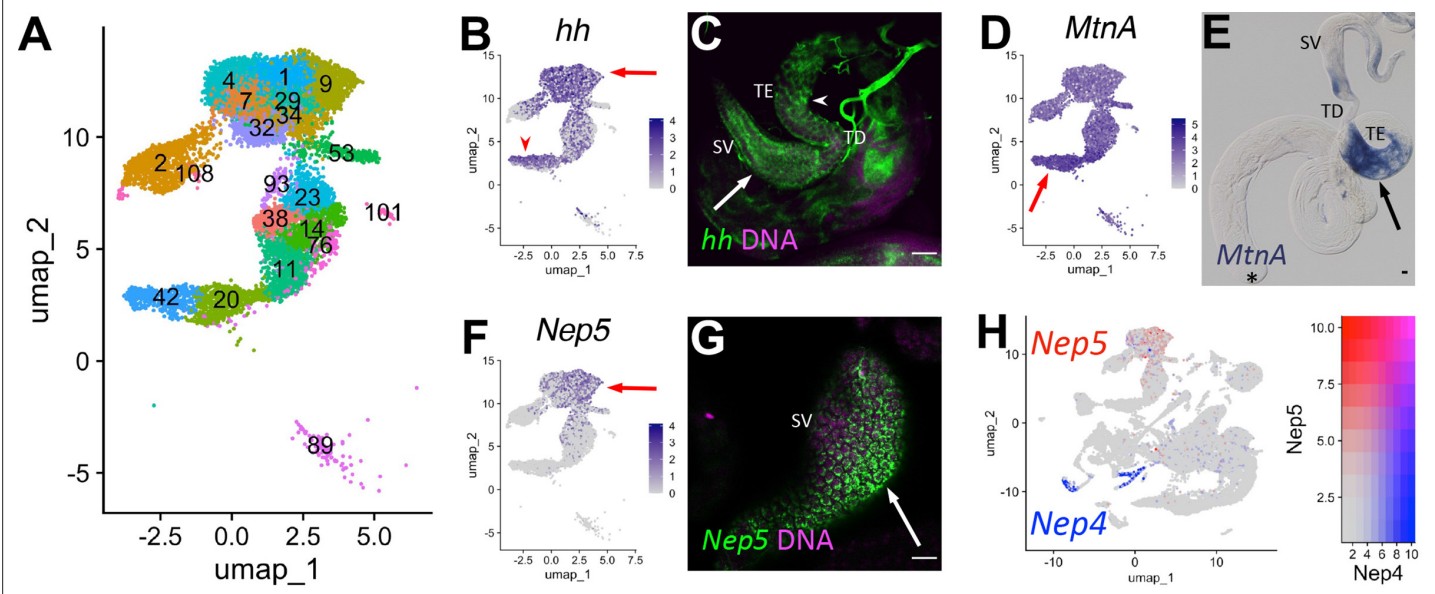

**Figure 8.** Characteristics of supporting epithelia. (**A**) UMAP of non cyst cell lineage epithelial cells of the testis from the FCA snRNA-seq data with Leiden 6.0 clusters. Note: identity of cluster 89 remains undefined. (**B - G**) UMAPs compared to ISH images. For each comparison, arrows and/or arrowheads point out the same cell type in the UMAP and its corresponding ISH image. (**B**) *hh* mRNA expression projected on the UMAP. (**C**) FISH of *hh* mRNA (green) and DNA (magenta) showing the base of the testis including the TE (arrowhead), testicular duct (TD), and SV (arrow). (**D**) *MtnA* mRNA expression projected on the UMAP. (**E**) Colorimetric in situ hybridization of *MtnA* mRNA (blue) in an entire testis plus seminal vesicle. Asterisk denotes hub. (**F**) *Nep5* mRNA expression projected on the UMAP. (**G**) FISH of *Nep5* mRNA (green) and DNA (magenta) showing the SV. (**H**) *Nep4* (blue) and *Nep5* (red) mRNA expression projected on the UMAP with corresponding heatmap. Arrows and arrowheads point to corresponding cell types in UMAPs and stained tissues. Bars: 20µm. (see also *Figure 8—figure supplement 1*).

The online version of this article includes the following figure supplement(s) for figure 8:

**Figure supplement 1.** RNA expression in non cyst cell testis epithelia.

while FISH showed *hh* RNA in TE and SV cells (*Figure 8C*, arrowhead and arrow, respectively). *Metallothionein A* (*MtnA*) was most highly upregulated in the lower clusters enriched for *hh* (*Figure 8A, B and D*; *Figure 8—figure supplement 1B*; clusters 42, 20), with colorimetric ISH confirming strong expression of *MntA* in the TE (*Figure 8E*, arrow). The SV marker *Neprilysin-5* (*Nep5*; *Sitnik et al., 2014*), confirmed by FISH to be expressed in SV, was upregulated in the uppermost clusters (mermaid head) (*Figure 8A, F and G*, arrow; *Figure 8—figure supplement 1C*; clusters 1, 4, 7, 9, 29, 32, 34, 93). Although they encode nearly identical proteins, *Nep4* and *Nep5* mRNAs were expressed in different cell types of the adult testis (HCC/TCC and SV, respectively; *Figure 8H*). Note that while the TE and SV each map as a single-block cluster at lower clustering resolution (*Figure 2—figure supplement 1B*), the appearance of multiple clusters for each cell type at higher resolution suggests notable transcriptional heterogeneity within both cell types (*Figure 8A*). Thus, epithelial cell types in general represent an underexplored area in testis biology.

## Discussion

Study of the *Drosophila* testis has had great impact on reproductive biology and on understanding stem-cell-niche interactions, developmentally regulated cell cycles and cell-type-specific cellular morphogenesis. Against this foundation, examining single cell and single nuclear RNA-seq has revealed several notable features. First, in gene expression space, the stem cell-based lineages for the germ line and the supporting somatic cyst cells were geographically isolated from each other, reflecting their distinct lineage identity. Second, the individual cells within each stem cell lineage were essentially arranged in strings, due to progressive changes in gene expression during differentiation. Third, a UMAP for each lineage revealed complexities along its trajectory that reflect both known intricacies of development within each lineage, as well as previously unappreciated gene expression dynamics. Fourth, in contrast to the two stem cell lineages, terminally differentiated cells generally

clustered into discrete groups, as expected. Finally, the comparison of single cell and single nucleus datasets proved particularly revealing of dynamic developmental transitions in lineage differentiation.

## Germline and soma map to distinct gene expression domains

While *Drosophila* spermatogenesis relies on two separate stem cell lineages, the progeny of each lineage associate intimately with each other, cooperating at multiple points during differentiation to produce functional sperm. Despite their physical association, in gene expression space the two lineages lie well-separated. This is not a surprise, since the lineages are specified independently during embryogenesis in space and time, as well as by different gene regulatory circuits. Such separation between germline and supporting soma is also observed in the scRNA-seq from the *Drosophila* ovary and murine testis (*Green et al., 2018*; *Rust et al., 2020*).

Accurate pairing of each germline cluster with a somatic cluster representing its interacting partner could facilitate identification of the underlying cell signaling circuits that form the basis for cooperation between the lineages at various points along their differentiation trajectory. We are able to highlight some likely pairing assignments due to the extensive knowledge of *Drosophila* spermatogenesis and testis biology. For example, the CySC cluster (62) likely associates directly with germ cells in cluster 25, while cyst cell lineage clusters 36 and 58 likely associate with later spermatogonia (cluster 22). Other associations are also suggested by the data (*Figure 6D*) with more remaining to be defined. The transcriptome in these pairings can be mined to identify candidate signaling pathways by recently developed tools, such as FlyPhone (*Liu et al., 2022*).

## Stem cell lineages appear as strings along their differentiation path

The data highlight how tissues maintained by stem cell lineages display a characteristic geography in gene expression space. The UMAP reveals these lineages to be arranged largely in strings, due to progressive changes in gene expression. The linear arrangement is strongly supported by trajectory inference for both the sc- and sn-RNA-seq datasets (see also *Li et al., 2022*). This arrangement might be diagnostic of at least some stem cell lineages, as it is observed to a degree in germline data from murine testes, in murine small intestine (*Green et al., 2018*; *Haber et al., 2017*), and in the follicle cell lineage in the *Drosophila* ovary (*Li et al., 2022*; *Rust et al., 2020*). While in some tissues maintained by stem cells, such an arrangement is made apparent only by trajectory inference, in the *Drosophila* testis it is apparent in the UMAP representing gene expression space.

Additionally, even with progressive changes in gene expression along their respective trajectories (*Figures 2L and 6U*), cells in these strings nevertheless can be sorted into clusters. The extensive study of germline and somatic testis biology provided excellent markers, which allowed assignment of cell stage identity with high confidence for many of these clusters. These assignments provide abundant opportunities to identify new, stage-specific markers to test for function in each stem cell lineage in vivo.

## Each stem cell lineage exhibited complexities, revealing known and previously unappreciated expression dynamics

In the testis, differentiation proceeds stepwise from the apical tip of the tubule. It is tempting to equate this progression with the arrangement of cell states revealed in the UMAP. However, the UMAP represents gene expression space and not physical space. This means that a stage characterized by dramatic changes in transcription in a differentiating lineage, while localized to a very small region within the organ, might be spread out in a thin stream in the UMAP. An example is seen in early spermatocytes, from completion of premeiotic S phase through to apolar spermatocytes (*Fuller, 1993*). Physically located in a narrow band next to the spermatogonial region in the testes, nuclei at these stages extend across much of the lower region of the germ cell UMAP (*Figure 2A*). A reciprocal case is highlighted by early spermatids, which have very low levels of transcription and thus show very low UMI values. These cells are clustered together in the UMAP, even though the cells are undergoing dramatic changes in morphology easily visible by microscopy (*Fuller, 1993*).

For the soma, the complexity observed within portions of the cyst lineage was a surprise. Head and tail cyst cells execute very different roles for the spermatid bundles with which they associate. Not surprisingly, the distinction between head and tail cyst cells is clear late in the differentiation process in the UMAP, when the two cell types have very different morphologies. Our analysis confirmed that

both head and tail cyst cells are derived from the same progenitor population (*Figure 6D*). However, the somatic clusters show complex intertwining before they eventually resolve to generate the two lineages. Interestingly, this complex tangling seems to coincide with the stages of spermatid development where there already is clear polarization in the architecture of the bundle (*Figure 6D*). Only by reprojecting this lineage, purposefully preserving an extra dimension, could we infer the trajectory properly, discerning a split and then a merge in expression profiles that tracked with cell identity. The initial split in the cyst lineage, as well as the subsequent and transient merge followed by a new split, suggests an interesting sequence of transcriptional cell states within this key supporting lineage.

## Differentiated cells map as discrete groups

Whereas the two stem cell lineages each appear as strings reflecting progression in their gene expression patterns, terminally differentiated cell types appear as more discrete patches in the UMAP (*Figure 1G*). This was the case for differentiated cell types that are an integral part of the testis, such as the hub and terminal epithelia, as well as for cell types that associate with the testis but are structurally distinct, such as the seminal vesicle. Even within a patch, however, increasing cluster resolution reveals complexity in cell type identity. For example, the snRNA-seq identifies a 'muscle' group (*Figure 1B*, 'mc') that is composed of several different Leiden 6.0 clusters (*Figure 2—figure supplement 1*, compare A with B). Perhaps the clusters reflect different muscle cell types, for example those covering the testis tubule vs. the seminal vesicle (*Susic-Jung et al., 2012*). Likewise, different clusters that comprise 'seminal vesicle' may represent distinct portions of the structure, such as entry and exit points (*Figure 8A*).

## The value of comparing sn- versus sc-RNA-seq

When carried out on the same tissue, single nucleus RNA-seq is typically comparable to single cell RNA-seq, with very high percent similarity in gene identification (*McLaughlin et al., 2021*). In the *Drosophila* testis, this was largely the case for germ cells early in the differentiation lineage (*Figure 4G and H*). In contrast, comparing these two approaches for later stage germ cells revealed a striking difference, reflecting an important aspect of testis biology. Our data show that directly comparing sn- to scRNA-seq can highlight cases where mRNAs are expressed at an early developmental stage (e.g. in spermatocytes) then stored for later use (e.g. in spermatids, where mRNAs may be used temporally). This phenomenon was previously described for a relatively small set of transcripts that persist in the cytoplasm for days following the cessation of transcriptional activity during *Drosophila* male meiosis (*Fingerhut et al., 2019*; *Kuhn et al., 1991*; *White-Cooper et al., 1998*). Our results demonstrate that such perdurance is likely a feature of hundreds of transcripts.

Key to this finding was the use of sn- and scRNA-seq datasets in tandem, rather than individually. snRNA-seq likely reports actively transcribed mRNAs at a given developmental time and therefore may be more sensitive to dynamic changes in transcription than scRNA-seq. Thus, snRNA-seq may be a better approach for mapping changes in cell state, for example, during embryonic development or differentiation in stem cell lineages. Data from scRNA-seq, by contrast, may be strongly influenced by mRNAs perduring in the cytoplasm from earlier time points, as well as mRNAs no longer being actively synthesized but purposefully stored for later usage. Thus, while snRNA-seq reveals gene expression dynamics through *active* transcriptional changes, scRNA-seq can capture post-transcriptional gene regulation, as is required by transcriptionally silent cells like early spermatids. Together, these datasets allow for inference of the full lifetime and velocity of all RNAs, from nuclear transcription to cytoplasmic degradation and/or protection.

Dynamic inference of transcript state by separately assaying nuclear and cytoplasmic RNAs, as used here, could prove a relevant measurement in many other differentiation systems as well. Cytoplasmic storage of transcripts expressed at an earlier stage to be used later is a widespread differentiation strategy and is especially predominant during oogenesis and spermatogenesis. Many mRNAs expressed during oogenesis are stored in the cytoplasm in a translationally silent state, to be recruited for translation in the early embryo (*Jenkins et al., 1978*). This is especially important in organisms with large, yolk-rich eggs, in which transcription from the zygotic genome is delayed until after several rounds of mitotic divisions. In the male germ line as well, where transcription ceases during spermiogenesis as the nucleus compacts, many mRNAs expressed at earlier stages are stored in the cytoplasm, initially translationally repressed, then recruited for translation during spermatid

morphogenesis. In both cases, perdurance of mRNAs in the cytoplasm after transcription has shut down is important to allow subsequent stages of development and differentiation to take place, and recruitment of specific mRNAs for translation at different times may play an important role in temporal control of morphogenetic events.

It is also notable that just 18/162 of the spermatid transcribed genes expressed late in spermio-genesis encode proteins detected in the mature sperm proteome (*Wasbrough et al., 2010*). While this might be due to limited sensitivity in proteome detection, alternatively, many may play roles in spermatid development but not mature sperm function. Examples could include regulating or medi-ating spermatid elongation, the histone-to-protamine transition, or individualization, as is the case for *soti* (*Barreau et al., 2008*; *Kaplan et al., 2010*). A further 20 of the 22 cytoskeletal or motor-related genes transcribed in spermatids (see *Figure 5—source data 1* for functional classes of the spermatid transcribed genes) have predicted functions in microtubule assembly, flagellar axoneme assembly, axonemal dynein regulation or microtubule transport, consistent with roles in assembling and elon-gating the 1.8 mm sperm flagellum and transporting cargos within this very long cell (*Ghosh-Roy et al., 2004*; *Noguchi et al., 2011*; *Tokuyasu, 1975*). The set of late upregulated genes identified as involved in lipid synthesis, lipid transfer and membrane trafficking could contribute to membrane addition at the distal (growing) ends of spermatid flagella to facilitate cell elongation (*Ghosh-Roy et al., 2004*). Finally, genes encoding RNA binding proteins could regulate transcript localization or translation in the increasingly long spermatid cells, as polarized mRNA localization by RNA binding proteins such as Orb2 (localizing *aPKC* and *orb2* mRNA) and Reptin and Pontin (localizing axonemal dynein mRNA) has been observed and shown to be important for sperm maturation (*Fingerhut and Yamashita, 2020*; *Xu et al., 2014*). Whether and how many other mRNAs are localized to the growing flagellar tip in spermatids remains to be studied.

## Use as a resource

With the transcriptional profiling of over 44,000 nuclei isolated from testis and associated supporting tissues, we have connected differentiation events throughout the germline and somatic lineages, capitalizing on the extensive literature on *Drosophila* testis biology. The expectation is that this will be a foundational resource for the field. Several other *Drosophila* RNA-seq and scRNA-seq efforts have been reported (*Gan et al., 2010*; *Hof-Michel and Bökel, 2020*; *Mahadevaraju et al., 2021*; *Shi et al., 2020*; *Vedelek et al., 2018*; *Witt et al., 2019*). Since each approach has different relative strengths and limitations, the foundation we have laid with the scRNA-seq and snRNA-seq datasets described here should assist others in comparisons with more stage-restricted transcriptome analyses. More broadly, the data presented here, in their easily shared formats, should enable a deeper exploration of the conserved aspects of germline and support cell biology during *Drosophila* and mammalian spermatogenesis.

## Materials and methods

**Key resources table**

| Reagent type (species) or resource | Designation | Source or reference | Identifiers | Additional information |
|---|---|---|---|---|
| Genetic reagent (*D. melanogaster*; male) | w1118 | BDSC | | |
| Genetic reagent (*D. melanogaster*; male) | Oregon-R | BDSC | | |
| Genetic reagent (*D. melanogaster*; male) | y1w1 | BDSC | | |
| Genetic reagent (*D. melanogaster*; male) | tup::GFP | BDSC | RRID:BDSC_81278 | y w; PBac{y[+mDint2] w[+mC]=tup GFP.FPTB}VK00031/ TM6C, Sb |
| Genetic reagent (*D. melanogaster*; male) | Amph::GFP | Kyoto Stock Center | CPTI-002789 | |

*Continued on next page*

*Continued*

| Reagent type (species) or resource | Designation | Source or reference | Identifiers | Additional information |
|---|---|---|---|---|
| Genetic reagent (*D. melanogaster*; male) | piwi::GFP | Kyoto Stock Center | CPTI-003588 | |
| Genetic reagent (*D. melanogaster*; male) | Akr1B::GFP | Kyoto Stock Center | CPTI-002728 | |
| Genetic reagent (*D. melanogaster*; male) | CG3902::GFP | Kyoto Stock Center | CPTI-100001 | |
| Genetic reagent (*D. melanogaster*; male) | *comr z2-1340* | White-Cooper | FBal0144389 | |
| Antibody | Anti-Fasciclin III, mouse monoclonal | Developmental Studies Hybridoma Bank | DSHB:7G10; RRID:AB_528238 | 1:50 |
| Antibody | Chick polyclonal anti-GFP | Abcam | ab13970 | 1:10,000 |
| Antibody | Goat anti-chicken Alexafluor 488 | Invitrogen | A11039 | 1:200' |
| Antibody | Goat anti-mouse Alexafluor 568 | Invitrogen | A11004 | 1:100' |
| Antibody | Sheep Anti-FITC, horseradish peroxidase | Roche | 11426338910 | 1:2000 |
| Antibody | Sheep Anti-Digoxygenin, horseradish peroxidase | Roche | 11207733910 | 1:1500 |
| Antibody | Sheep Anti-digoxygenin, alkaline phosophatase | Roche | 11093274910 | 1:2000 |
| Antibody | mouse anti coracle monoclonal | Developmental Studies Hybridoma Bank | DSHB: C615.16 | 1:20 |
| Recombinant DNA reagent | piwi (plasmid containing cDNA) | BDGP DGC library | GM05853 | FBcl0142639 |
| Recombinant DNA reagent | eya (plasmid containing cDNA) | BDGP DGC library | GH05272 | FBcl0108545 |
| Recombinant DNA reagent | f-cup (plasmid containing cDNA) | BDGP DGC library | GH09045 | FBcl0128895 |
| Recombinant DNA reagent | loqs (plasmid containing cDNA) | BDGP DGC library | RE14437 | FBcl0204474 |
| Recombinant DNA reagent | geko (plasmid containing cDNA) | BDGP DGC library | RE30284 | FBcl0192532 |
| Recombinant DNA reagent | Vsx1 (plasmid containing cDNA) | BDGP DGC library | SD01032 | FBcl0286608 |
| Chemical compound, drug | DAPI | Millipore Sigma | 1023627600 | 1:1000 (of 2–5 mg/ml) |
| Chemical compound, drug | Vectashield | Vector Labs | H-1000 | |
| Chemical compound, drug | GMM tissue clearing reagent | Gary Struhl | | 2 g Canada Balsam in 1 ml methyl salicylate |
| Chemical compound, drug | Rhodamine tyramide | Thermo Scientific | 46406 | 1:1000 |
| Chemical compound, drug | Fluorescein tyramide | Thermo Scientific | 46410 | 1:1500 |
| Commercial assay or kit | Stellaris RNA FISH Probe Designer | Biosearch Technologies Inc. | | |

*Continued on next page*

*Continued*

| Reagent type (species) or resource | Designation | Source or reference | Identifiers | Additional information |
|---|---|---|---|---|
| Commercial assay or kit | 10 X Genomics Chromium Controller; Next GEM Single Cell3' v3.1 | https://www.10xgenomics.com/product-catalog | 1000269 | |
| Software, algorithm | Seurat 4.0.5 | *Satija et al., 2015*; *Hao et al., 2015* | | |
| Software, algorithm | Monocle3 | https://cole-trapnell-lab.github.io/monocle3/; *Trapnell et al., 2014* | | |
| Software, algorithm | Venny 2.1 | https://bioinfogp.cnb.csic.es/tools.html | | |
| Other | Olympus Bx50 microscope | Olympus | | |
| Other | 20 x/0.60 NA objective | Olympus | | |
| Other | 10 x/0.30 NA objective | Olympus | | |
| Other | 40 x/0.75 NA objective | Olympus | | |
| Other | KY F75U Camera | JVC | | |
| Other | Orca-05G Camera | Hamamatsuu | | |
| Other | 63 x/1.4 NA objective | Zeiss | | |
| Other | LSM800 confocal | Zeiss | | |
| Other | 63 x/1.4 NA objective | Leica | | |
| Other | Stellaris Confocal | Leica | | |

## *Drosophila* lines

For snRNA-seq, testes and attached seminal vesicles were dissected from 0 to 1-day-old $w^{1118}$ males and processed as described in *Li et al., 2022*. For scRNA-seq, testes alone were dissected from 1 to 5-day-old Oregon-R males. Oregon-R testes were also used for in situ hybridization of *aub* and *fzo* (*Figure 2*). $y^1w^1$ flies were used for smFISH in *Figure 5*. Amph::GFP (CPTI-002789), piwi::GFP (CPTI-003588), Akr1B::GFP (CPTI-002728), CG3902::GFP (CPTI-100001) and Tup::GFP (BDSC line 81278) were from CPTI and Bac collections (*Kudron et al., 2018*; *Lowe et al., 2014*). $comr^{z2-1340}$ homozygotes and $w^{1118}$ were used for colorimetric in situ hybridization.

## Testis squashes and analysis of expression of fluorescent fusion proteins

All tissue analyses included samples from multiple males of each genotype (5–10 pairs of testes per experiment), typically processed as a single batch. Testes from transgenic flies of the YFP CPTI collections were dissected in testis buffer (183 mM KCl, 47 mM NaCl, 10 mM Tris pH6.8), cut open using tweezers and gently squashed on a glass slide by application of a coverslip. Testis squashed preparations were imaged live in sequentially captured images by phase contrast and epifluorescence microscopy using an Olympus Bx50 microscope, with 20 x, 0.60 NA, 10 x, 0.30 NA, or 40 x, 0.75 NA UPlanFl objectives and either a JVC KY F75U or a Hamamatsu Orca-05G camera.

For Tup::GFP analysis, testes were dissected from BAC transgenic flies, fixed in 4% paraformaldehyde for 20 min, and blocked in 3% bovine serum albumin, 0.02% sodium azide, 0.1% Triton X-100 in phosphate-buffered saline for 1 hr. After a 1 hr wash with PBX (0.1% Triton X-100 in 1 X phosphate-buffered saline, pH 7.4), testes were incubated overnight at 4 °C with antibodies to GFP (Abcam 13970; 1:10,000) and Fasciclin-III (Developmental Studies Hybridoma Bank 7G10; 1:50). After washing with PBX testes were treated with goat anti-chicken AlexaFluor 488 (Invitrogen A, 11039; 1:200), goat anti-mouse AlexaFluor 568 (Invitrogen A11004; 1:100), and DAPI (Millipore Sigma 1023627600; 1:1000) for 2 hr. After final washes in PBX, testes were mounted in VectaShield (Vector Labs H-1000), and images were captured at 63 X, NA 1.4, on a Zeiss LSM800 confocal microscope.

## RNA in situ hybridization

For in situs presented in *Figures 2 and 4*: *aub* (forward primer 5'-CCTGGGCGGCTACATCTT-3'; reverse primer 5'-GCGCAGATTTCGACTCGG'–3), *kmg* (forward primer 5'-TGCCTCTATGCCTCAC GC-3'; reverse primer 5'- GCGCCTACCGGTCTCATC-3"), *fzo* (forward primer 5'-GGCATCCAAACT CTCGCG-3'; reverse primer 5'-TGTCGCAACTGGAGCTCA-3') were amplified by PCR on cDNA from Oregon-R *Drosophila* Testes. Using TA cloning (Promega, 'Easy-T' Cloning), the resulting amplicons were cloned into the pGEM vector (Promega). Subsequent PCR added a T7 binding site (5'-GAAG TAATACGACTCACTATAGGGAGAGGG-3') upstream of the amplicon. The resulting plasmids then served as templates for in vitro transcription with Digoxigenin (DIG)- and fluorescein isothiocynate (FITC)-labeled ribonucleotides to generate labeled single-stranded antisense riboprobes.

Testes were isolated and fixed in 4% formaldehyde for 30–60 min, dehydrated into methanol and stored at –20 °C for up to 1 month. After rehydration in PBS +0.1% Triton-X100, testes were permeabilized with 4 µg/ml Proteinase K for 6 min and washed with Pre-Hybridization solution (50% deionized formamide, 5 x SSC, 1 mg/ml yeast RNA, 1% Tween-20) for up to 2 hr at 56 °C. Testes were incubated overnight at 56 °C with probes diluted 1:800 in Hybridization solution (50% deionized formamide, 5 x SSC, 1 mg/mL yeast RNA, 1% Tween-20, and 5% Dextran Sulfate). After washes in Pre-Hybridization solution, 2 x SSC, and 0.2 x SSC, then PBS +0.1% Triton-X100, samples were blocked for 30 min in 1% Roche Western Blocking Reagent prior to incubating overnight at 4 °C with either anti-FITC with horseradish peroxidase conjugate (Roche) at a 1:2000 concentration or anti-DIG- with horseradish peroxidase conjugate (Roche) at a 1:1500 concentration in 1% Roche Western Blocking Reagent. Fluorescent tyramide development and amplification were performed by first placing the testes for 5 min in borate buffer (0.1 M boric acid, 2 M NaCl, pH 8.5), followed by 10 min in borate buffer with rhodamine (1:1000) or fluorescein (1:1500) tyramide, and 0.0003% hydrogen peroxide. After development, peroxidase activation was performed in a 1% sodium azide solution for at least 1 hr, followed by antibody labeling for the second probe. Coverslips were mounted in Vectashield with DAPI (Vector Labs). Fluorescence image acquisition was performed on a Leica Stellaris Confocal microscope using a 63 X oil-immersion objective (NA = 1.4).

## For in situs presented in Figures 5, 6, 8, and Figure 3 – figure supplement 2

Fixation and hybridization for FISH was as described for *Figures 2 and 4*, with minor modifications as described (*Wilk et al., 2017*). Briefly, these include using a cold acetone permeabilization step and 0.3% Triton X-100 instead of Proteinase K for improved tissue permeabilization, and DIG-labeled probe detection via tyramide amplification. DAPI (4",6-diamidino-2-phenylindole; Sigma, cat. no. D-9542) was used at 1 µg/ml to reveal nuclei. For detection of *piwi*, *eya*, *f-cup*, *loqs*, *geko,* and *Vsx1* transcripts, RNA probes were transcribed from the BDGP DGC library plasmid clones GM05853, GH05272, GH09045, RE14437, RE30284, and SD01032, respectively using T7 RNA polymerase. Testes were incubated with an antibody to Coracle (Developmental Studies Hybridoma Bank C615.16; 1:20) for detection of Coracle protein in *Figure 3—figure supplement 2*. For *hh* and *Nep5*, templates were made by PCR from genomic DNA using the following T7 and T3 promoter-containing primers:

> *Hh* last exon:
> Forward: GTAATACGACTCACTATAGGGAGACCACTGCCGATTGATTTTCTCAGG
> Reverse: AATTAACCCTCACTAAAGGGTTGTGGAGATCGTGTTTTGAGCAT
> *Nep5* exon 6:
> Forward: GTAATACGACTCACTATAGGGAGACCACGGGGAAATCCGATAAAGCTC
> Reverse: AATTAACCCTCACTAAAGGGTTGTATCTGCAGAACCAAACTGAC

## For colorimetric in situ hybridisation in Figure 3, 8 and Figure 3 - figure supplement 2

Probe preparation and in situ hybridisation were performed as described (*Morris et al., 2009*). Primers Hml-F ATTTAGGTGACACTATAGAATAAGTGGACCCATGCCAAG and Hml-R TAACCCTC ACTAAAGGGTGACCATCATCGCAAATC and. primers MtnA-F ATTTAGGTGACACTATAGAAGCGG TAAGTTCGCAGTC and MtnA-R TAACCCTCACTAAAGGGACATTTATTGCAGGGTGTG were used to amplify 628 bp and 443 bp fragments respectively from cDNA generated from *w*[1118] testes.

After re-amplification using primers 5'-SP6 ACGGCAATTTAGGTGACACTATAGAA and 3'-T3 GCAA CGAATTAACCCTCACTAAAGGG, and the products served as templates for T3 RNA polymerase to generate dig-labeled single-stranded, antisense RNA probes. The probes were hydrolysed for 15 min, precipitated and resuspended in 200 µl water. Testes were dissected from young $w^{1118}$ males, fixed in 4% paraformaldehyde for 20–60 min, washed in PBS, permeabilised with 50 µg/ml proteinase K for 5 min, washed in PBS, then hybridisation buffer (HB: 50% Formamide, 5×SSC, 100 µg/ml denatured sonicated salmon sperm DNA, 50 µg/ml heparin, 0.1% Tween 20, 100 mM citric acid). Probes were diluted 1:100 in HB and testes were hybridized for 16 hr at 65 °C. Testes were washed 6x30 min at 65 °C in HB, followed by 15 min each step at room temp in 4:1, 3:2, 2:3 1:4 HB:PBST, 2x15 min PBST, then incubated overnight at 4 °C in alkaline phosphatase-conjugated anti-digoxygenin antibody diluted 1:2,000 in PBST. Testes were washed 4x15 min in PBST and finally 3x5 min HP (100 mM NaCl, 100 mM Tris (pH 9.5), 50 mM MgCl2, 0.1% Tween 20). NBT and BCIP diluted in HP were added as a colorimetric substrate for alkaline phosphatase and color allowed to develop in the dark at room temperature. The testes were washed in PBST, dehydrated through an ethanol series, mounted in GMM, and imaged using DIC microscopy (10 x objective) with a JVC KY F75U camera mounted on an Olympus BX50 microscope.

## For single molecule FISH presented in Figure 4

Testes were dissected in 1 X PBS and fixed in 4% formaldehyde in 1 X PBS for 20 min. Testes were then washed twice for 5 min each in 1 X PBS with 0.1% Tween 20 followed by permeabilization in 1 X PBS with 0.1% Triton-X 100 for 2 hr. Next, the testes were washed twice with 5 X SSCT (0.1% Tween 20) for 5 min each. Probes were added to pre-warmed Probe Hybridization Buffer (Molecular Instruments) to a final concentration of 16 nM. 100 µL of hybridization solution was added to each sample, pipetted to mix, and allowed to hybridize overnight at 37 C. Samples were then washed four times with Probe Wash Buffer (Molecular Instruments) for 15 min each at 37 C. During the first wash, amplifier hairpins (Molecular Instruments) were individually incubated at 95 C for 90 s and allowed to slowly cool to room temperature. Hairpins were then added to Amplification Buffer (Molecular Instruments) to a concentration of 60 nM. After washing the samples twice with 5 C SSCT for 5 min each, 100 µL of hairpin solution was added and the samples were allowed to incubate at room temperature overnight. The testes were then washed twice with 5 X SSCT for 30 min each at room temperature, mounted in Vectashield with DAPI, and imaged on a Leica Stellaris 8 confocal microscope with a 63 X oil immersion objective lens (NA = 1.4) and processed using ImageJ and Adobe Photoshop software. Probes (sequences below) and amplifier hairpins designed by Molecular Instruments (https://www.molecularinstruments.com).

| l(2)41Ab exon probe-binding sequences | l(2)41Ab intron probe-binding sequences |
| --- | --- |
| CATgAAAgAATggCAgAACTAggggACATgCAAAgCTTAggAAgTAACATCA | CTCTCgCTgTCTgCAAgCgAATAAAAACgATTACTCTATAACATgCATgTgA |
| CTATCAATgCTTgAggTTgAAAAACTTAgATgCgTAgTTTTTAATgAAAgAg | TCgACggACATTCCAgCATCTATCATCTTCTTTACTgATAgCAAATTgTgCg |
| ggAgAATTgCAAgTTCgTgAACATCAgCAgTTTTggATAATgCTgCAgAACC | gCCAACCTCTCATgCCACAAgCgATACTCTgCATCATTTgATATTgCAgATg |
| CgAAAACAACTATCCATTATTAggCAAAAgTCATTAAAAAgTgACCAAgAgC | AgCgTAATCggACCgCATACATCCgATTgAATTACAAATAgTgggCgTTCAA |
| AgAgACgCTCAgTTgCAAgACCAgCTATCACAggAAAAACgAAAATCAgATA | TTAgTgCTTATACTgTAgTTgTgggTATTAggAACCgCTAAATgATAgCTgT |
| ACACAggCATggAACgAggTTATTATAgCCATTCAgCTgAgAgATAAAATTT | CgACCAgTATAgTTTTgATCgATTTCTCgTCAAATTTTCgCTTTCgAACTTT |
| TTTCAAgTACAAgACgCAgCAAgAAAACACAgggAggCAATgAAACACAATA | TCATAAATTTTCTCgAgTTCTCATTCCACAACTTATAgCCATTTggCTCATA |
| AAAATTggATTCCTTTAAgCAACgACTggCAgATgAAACTAgTTCCTTAAgAA | CATTTgAAgTCACCgATTTCTTgTTTATTTCATCCACCACTACATCTCTAgC |
| gAATgTCAggATgAAAgCgAATCTCTTTTgAAgATgCAAgCAAAgCTAgATg | TTAgAgAATCACTgTATTggCCAAAAAAATCATgTTTTTCCACAgAgTCATT |
| ggAAgTCATCTATTCACAgAgTTCgAAAACTAgACATCgCCTTggATAAAT | TCgTAgATTCACAATTTTCCTgCAATggAAAATTCgACTCTTCATTCTCTTT |
| TTAACgCAAAAgTTCCgCACTgAggAAACAATTCTACggTTAgCgCAggATC | TgTATTCTCgAgACACgTggAATTTTCATTTggATTTTCATCAgACTTCgAA |
| TTAATAACAgACAAggCTACCgCTTACCgTTTgAAACTATTAggTAAATCTC | gATTTTAATggCATCCTCCCACAAggTTCTgTCTTTACgAgTTTTAATCTCT |
| gTAgAggAgACTgAgTTCgACCACgAACAggCTgAgCAAACTCAAAAACTAA | gATCggCTCgAAgTgAgTCgCgAgAAACgAAAgTgTCTAgCCAgCATATTTg |
| ATTgCggAgCTggAAAATACTTAAATTggAAggAAAAATTgTTgAgCTACAAA | CgACTTAAgATACTTATCgATgTACTTAAgAggAggAgTCCCCCAACTggAT |
| CgCTCTgTAAAAggTgAAATCggTCTTATgAAAACTgAAATACACCgCATgA | ACTTTCAAATTCCTTTTAAATgATAACCTTAAATTgACAgTTgCCTTTAgAT |

*Continued on next page*

*Continued*

| l(2)41Ab exon probe-binding sequences | l(2)41Ab intron probe-binding sequences |
|---|---|
| CATTgCgTTATgCATCgTgAACAAATCTTTgTCAATgCATCggTCAAggAgC | TATTCAgCTTCggTAgATgAAgCAgCAACCgATTTTTgTTTAATAgTACTCC |
| CgTAgAATTCAgTCAgATCTTAACgATATCATTAAggAggACgCCAATATTC | gTTggTAgTggCTTTgTCAAAATATTAgAAAATTgTAAgTCTgTggAAAgAT |
| gATCgAATAgAAgAAggAATATTAgCTAAgCATgCCAACTTggAACAAATAA | ATCATTCAAgATTCggTTAgCCCCAgTTggTCTCgCAgTgCAgTgAATCTgg |
| CAAgTAATACTCCgTTCAgAgTTgACAgTTCAACAgCTATCTCAAAAgCAAA | ATTTCTTTACAATACAACACgTTgTgTTATgTAATgTTgCTTCCAgAATggT |
| gAgCTAAATTATgATCTCACAgAAATAgCgAAggCTCTTTTACTAgACTATC | AgTTTAACTTgTCCTTATgCAgTAgCATAAACAAATTCTCCACgCTTTgCAA |

## For single molecule FISH presented in Figure 5

smFISH was performed as previously described (*Fingerhut et al., 2019*). Briefly, testes were dissected in 1xPBS (Invitrogen, AM9624) and fixed in 4% formaldehyde (Polysciences, Inc, 18814–10) in 1xPBS for 30 min, washed twice in 1xPBS for 5 min each, and permeabilized in 70% ethanol overnight at 4 °C. Testes were washed with wash buffer (2 x saline-sodium citrate [SSC, Invitrogen, AM9770], 10% formamide [Fisher Scientific, BP227]), and then hybridized overnight at 37 °C in hybridization buffer (2xSSC, 10% dextran sulfate [Sigma-Aldrich, D8906], 1 mg/mL yeast tRNA [Sigma-Aldrich, R8759], 2 mM Vanadyl Ribonucleoside complex [New England Biolabs, S1402S], 0.5% BSA [Invitrogen, Am2616], 10% formamide). Following hybridization, samples were washed twice with wash buffer for 30 min each at 37 °C and mounted in Vectashield with DAPI (Vector Laboratory, H-1200, Burlingame, USA). Images were acquired using an upright Leica Stellaris 8 confocal microscope with a 63 X oil immersion objective lens (NA = 1.4) and processed using ImageJ and Adobe Photoshop software (National Institutes of Health, Bethesda, USA).

Fluorescently labeled probes were added to the hybridization buffer to a final concentration of 100 nM. Probes against *soti* and *wa-cup* transcripts were designed using the Stellaris RNA FISH Probe Designer (Biosearch Technologies, Inc, Novato, USA) available online at https://www.biosearchtech.com/stellarisdesigner/.

| Probe Target | Fluorophore | 5'-Sequence-3' |
|---|---|---|
| *wa-cup* | Quasar 670 | cttccgataagctcatttgg, ccgctgttgggtgaaaaaga, ccagagcgctgttgaaatac, gagctctttcattgaacgga, atcgatcttttcagctgact, ctttcggctcatcaacagat, gactgacattgggattggtt, tacagagcatcgcagacttc, gaatcttccaagcgattgga, ccggagctaaatcgctttaa, tgagacggtcgagaacagga, gacatggtggtatcatctga, aactagccatcatgcgattg, gtatccttaatatccttggg, gttttccattatgctaacca, agtggggacaaatgggtttc, ttaaaatgccttttttcgcct, ccagcatttgttcagatacg, tcgcagtgtcttcagaaagg, gacttttttgcaatgcttggg, catcagtttgcccaaatact, tgcgttccgctttataattg, taaggcgtacatgggacttc, ctcgttgggatatttctgtt, ctcccggtgcttattataaa, gtctggtactgaatgcgata, atgctgcgcagaatcttgaa, gttctccaactcgaattagc, cccatgacttcctcaataaa, cgtcttgatggtgacatagt, aaatttccacggcattacgc, aatgcgagctaaacccaagt, agacagtcatattgctggga, gtgttagcagacgttgtttg, actcgttcgttttgtctttg, gtagccgatctggttatatt, ttaaaatgctccgctttggg, atacgattttccagtcggac, gttcaatgtgatactcggca, aattcgtgcagtagataggc, acaattcagatgctcttggg, ctccatataacactcttgca, gccttgcatataaccatgag, taagcacaggtcaaggttct, ctcctccgcattaactttaa, tgagccaaacttttgtctct, ctgatcgttgctttggaaca, taatttggttgcgatcctca |
| *soti* | Quasar 570 | tctcgacgaggtaatttg, tccgtgtagtacgtccat, gctcatcgtacagatcgt, ccgactcgatcgattagc, atcttcattcaccgcgtc, tgtccaagtcatcgccag, tgctgtccatcctccaat, tgacgattgactcccagg, gtccaggagtatgtccat, caacggtggctcttgagg, ctccttgcgccggaaaaa, acgtggtggtccatttgg, aacttcgtttcttccgcc, ggagtgggtttggtcata, ctcctgactttggcatgg, ttaggaggcacatctccg, attgccctcgtgacactg, atcctcgcgaacgtgacg, caaagtactcgcctcgct, gggtagttctgactggtc, tggcagacccataccatt, agaactgaccccaatgct |

## Preparation of samples for snRNA-seq

FACS-sorted nuclei were obtained from hand dissected, 0- to 1-day-old adult testes (plus seminal vesicles) and processed as described by the Fly Cell Atlas project (*Li et al., 2022*). Data from 44,621 sequenced nuclei passed quality control metrics. The raw data are publicly available (https://www.ebi.ac.uk/arrayexpress/experiments/E-MTAB-10519/).

## Importing FCA data into Seurat

The 'Testis, 10 X, Relaxed' loom file (https://flycellatlas.org/; https://cloud.flycellatlas.org/index.php/s/XAq3kCkfwEQotw7; *Li et al., 2022*) was imported into Seurat 4.0.5, and a standard pipeline run on the resulting Seurat R object to normalize and scale the data (NormalizeData, FindVariable-Features, and ScaleData). The loom file had already been filtered with nuclei expressing less than 200 genes or exceeding 15% mitochondrial content removed and genes not expressed in at least three nuclei removed. We chose to use the relaxed rather than a '10 X, Stringent' testis loom file due to the particular biology of the testis. Testis germline cells can express 'somatic' genes (see Results), including the hub cell marker *upd1*. As a consequence, filtering algorithms that generated the stringent dataset led to loss of a documented somatic cell type (the hub), as well as a large number of late spermatocytes. The Fly Cell Atlas website provides links and tutorials to ASAP and SCope, two web-based pipeline and visualization portals where users can examine the data or re-run analyses (Davie, 2018; Gardeux, 2017). Within the SCope interface, select the 10X>Testis > Relaxed dataset, and Settings >HVG UMAP. The analysis in this paper complements SCope and ASAP with Seurat and Monocle3, two R programming-based tools for single cell analysis (*Satija et al., 2015*; *Trapnell et al., 2014*, *Hao et al., 2015*; https://cole-trapnell-lab.github.io/monocle3/; *Pliner et al., 2022*).

The cluster information contained in the original loom file was preserved in the Seurat object, with clusterings available at increasing levels of resolution (Leiden algorithm, 0.4–10). The level of granularity provided by resolution 6.0 was deemed most revealing and is used for most analysis here. The full Seurat Object, FCAloomToSeurat2TFP_Annotations.rds, is available for download at https://doi.org/10.5061/dryad.m63xsj454, as is the script used to produce it from the loom file (convert_loom_to_seurat_normScale_adjHub.R).

Original cluster 90 was manually split into two in the Seurat Object, FCAloomToSeurat2TFP_Anno-tations.rds, with one resultant cluster of 79 nuclei definitively established as representing the hub (retaining its cluster number, #90), and the remaining nuclei placed into a new cluster, #111. Venn diagrams were created using Venny 2.1 (Oliveros, J.C. 2007–2015).

In some cases, data were extracted from subsets of the original Seurat object: a germline only subset (*Figures 2–5*), a somatic cyst cell lineage subset (*Figure 6*), and a subset representing several epithelial cell types of the testis as well as specific additional epithelial cell types (*Figure 8*). All UMAPs were generated within Seurat using the 'DimPlot' function. UMAPs that highlighted particular genes of interest were generated using the 'FeaturePlot' function (*Figures 1–3* and *5–8*). Dotplots were generated using the 'DotPlot' function (*Figures 2–7*).

The testis 10 X Relaxed loom file only contains a UMAP reduction projected to two dimensions. To inspect a 3D UMAP representation, the appropriate lineage was isolated and reprojected, passing the argument 'n.components=3 L' to the Seurat function 'RunUMAP' (see cystlineage3Dcode.R code; *Qadir et al., 2020*; 10.5281/zenodo.348317). An html version of the resulting 3D representation is available for download at https://doi.org/10.5061/dryad.m63xsj454.

For analysis of heart, Malpghian tubule and male reproductive gland (*Figure 3—figure supplement 1A*), the appropriate stringent 10 X loom file from Fly Cell Atlas was imported into Seurat. Means were calculated for nCount (UMI) and nFeature (gene). Cluster numbers were assigned using the 'Annotation' metadata field in each object, and a plot produced for average UMI by cluster number, with dot size reflecting average gene number per cluster.

Graphs generated directly from ASAP were used to produce *Figure 3D–E*; *Gardeux et al., 2017*. Continuous Coloring of a Custom Gene Set - Categorical Gene Metadata: _Chromosomes (either X or Y) was used under the Visualization tab. SVGs can be saved directly from the website. The enrichment analysis performed to produce these graphs mirrors Seurat's AddModuleScore function and is detailed in the Materials and Methods of *Li et al., 2022* (see header: Metabolic clustering using ASAP).

On occasion, use of an alternative Assay ('log.counts') in the Seurat Object allowed for visualization of low levels of gene expression in spermatocytes. This 'log.counts' assay contains a matrix of $\log_2(\text{counts} + 1)$, or log-transformed raw counts. This was done for plotting or to perform analyses focused on promiscuous expression in spermatocytes (*Figure 3F–G*; *Figure 3—figure supplement 2A–D, G1*; *Figure 3—source data 1*). *Figure 3* code describes how this assay was added and shows how to utilize this information when needed.

Cell-type-specific transcription factors were taken from *Li et al., 2022*, Table S3 and are available at https://doi.org/10.5061/dryad.m63xsj454 (TFs_list_500.txt). Each gene was classified as being scored as expressed only in mature spermatocytes, both in other cell types and mature spermatocytes, or not in germ line cell types as per assignments in the Fly Cell Atlas Table S3 (*Li et al., 2022*). *Figure 3—source data 1* was generated by calculating average log-transformed expression of each transcription factor per Leiden 0.4 cluster. Subtracting the value in cluster 3 from that of cluster 2 yielded the upregulation of each gene in mature spermatocytes relative to spermatogonia and early spermatocytes (*Figure 3—source data 1*). The DotPlot in *Figure 3I*, in contrast, utilized only the testis and seminal vesicle data to summarize the phenomenon seen with the Fly Cell Atlas data using just the nuclei derived from the dissected testis and seminal vesicle sample. A subset of clusters for the cell types of interest at Leiden 0.4, including clusters 2 and 3 as mentioned above, were used to generate the plot.

## Tissue isolation and cell dissociation for scRNA-seq

The testis dissociation protocol was adapted from *Witt et al., 2019*. Fresh maceration buffer (10 mL Trypsin LE (Gibco) with 20 mg collagenase (Gibco)) was prepared on the day of the dissection. Testes were hand dissected from 1 to 5-day-old male flies in 1 x phosphate-buffered saline (PBS), separated from seminal vesicles and transferred immediately into tubes filled with cold PBS, on ice. Testes were kept in PBS for a maximum of 30 min. Samples were centrifuged at 135 rcf and the PBS removed and replaced with 400 µL maceration buffer. Testes were incubated in maceration buffer for 30 min with gentle vortexing every 10 min at room temperature. Following incubation, samples were pipetted up and down for 15 min until all visible chunks were gone and the sample was in approximately a single-cell suspension. Sample was filtered through a 35 µm filter into a polystyrene tube, then transferred into a microcentrifuge tube. After the sample was centrifuged at 135 rcf for 7 min, the supernatant was removed and the pellet resuspended in 1 mL calcium- and magnesium-free Hanks' Balanced Salt Solution (HBSS). The sample was spun a final time at 137 rcf for 7 min. All but 50 µL of the HBSS supernatant was removed, and the cell pellet was resuspended in the remaining 50 µL. Cell viability and density was then assayed on a hemocytometer using DIC imaging and Trypan Blue stain.

## Library preparation and sequencing for scRNA-seq

Cells were processed using the 10 x Genomics Chromium Controller and Chromium Single Cell Library and Gel Bead Kit following standard manufacturer's protocol. Amplified cDNA libraries were quantified by bioanalyzer and size selected using AMPure beads. Samples were sequenced on a NovaSeq SP.

## Mapping and preprocessing of scRNA-seq data

Reads were mapped to the DM3 reference genome using the 10 X CellRanger pipeline with default parameters. The resulting feature matrix (default output, kept in outs/filtered_feature_bc_matrix, and featuring barcodes.tsv.gz, features.tsv.gz, and matrix.mtx.gz) was read into the R package Monocle3 using load_cellranger_data. The resulting cell data set (cds) object was processed using 100 dimensions, and underwent dimensionality reduction using the UMAP method. Germline cells were identified on the basis of super-cluster (in Monocle3, 'partition') identity, with 100 dimensions used to identify partitions. The germline cells were then subsequently clustered using the Monocle3 'cluster_cells' command, with resolution = 0.003.

## Trajectory analysis of FCA snRNA-seq data

The publicly available 'Testis, 10 X, Relaxed' loom file of snRNA-seq data from the Fly Cell Atlas website (https://flycellatlas.org/; 10.1101/2021.07.04.451050) was read into Monocle3, and preprocessed using 50 dimensions. This dimension number was determined empirically as it resulted in connected clusters that represented the primary lineages (germline and cyst). UMAP dimensionality reduction and clustering was performed with a resolution of 0.0002, again determined empirically to represent biologically significant clusters that approximated the original annotation. A trajectory graph was generated from data with 'learn_graph'. Pseudotime was calculated with 'order_cells', with the first (base) node selected as the root in Monocle3's interactive mode. Pseudotime parameters were then subsequently visualized on the original projection by adding a 'pseudotime' slot to the FCA

Seurat object. 'learn_graph' and 'order_cells' were likewise run on the scRNA-seq dataset, with the single most base node again selected as the root.

A list of genes that change expression level dynamically along pseudotime was generated from the full join of genes that vary along pseudotime in the single cell and nucleus datasets, according to the graph_test function in Monocle3 (parameters: neighbor_graph = 'principal_graph', method = 'Moran_I' cores = 4; selected genes with q_value = 0 and morans_I=0.25). Additionally, several genes encoding transcripts that were annotated as enriched in late pseudotime in the original FCA paper were added to the analysis. Genes (representing rows on the *Figure 4G–H* heatmaps) were ordered according to pseudotime point of peak expression averaged between the two datasets. A z-score for each gene expression for each dataset, smoothed across pseudotime using R function 'smooth.spline' with 3 degrees of freedom was calculated and the heatmap was generated using the R package ComplexHeatmap.

## Aligning trajectories for scRNA-seq and FCA snRNA-seq data by dynamic time warping

The trajectories for the scRNA-seq and snRNA-seq data were aligned on a common 'warped' time scale using a Dynamic Time Warping based procedure (adapted from *Alpert et al., 2018*; *Cacchiarelli et al., 2018*). The smoothened gene expression (as shown in the heatmaps) for all germline cells in each of the two monocle3 trajectories was used for alignment using the dtw function in R package dtw. The Pearson's correlation based distance and 'symmetric2' step parameters were used in dtw. The aligned time scale returned by dtw was used as the warped pseudotime. Transcript expression levels could then be plotted on the same axis.

## Acknowledgements

We especially acknowledge and thank the Fly Cell Atlas project, the Chan-Zuckerberg Biohub, the NIH sequencing facility and the laboratory of Steve Quake for the starting testis plus seminal vesicle snRNA-seq dataset. We thank Stein Aerts and Bart Deplancke for supporting the efforts of JJ and WS, respectively. We acknowledge the Whitehead Institute Genome Technology Core for assistance in library preparation and sequencing of the scRNA-seq dataset, the Bloomington *Drosophila* Stock Center and Kyoto DGGR for *Drosophila* strains, the Developmental Studies Hybridoma Bank for various antibodies, and Aaron Allen for advice and initial code to transfer the SCope loom file format into SEURAT. SD thanks Dan Beiting at PennVet for his DIY_Transcriptomics course.

## Additional information

### Competing interests

Yukiko M Yamashita: Reviewing editor, eLife. The other authors declare that no competing interests exist.

### Funding

| Funder | Grant reference number | Author |
| --- | --- | --- |
| National Institutes of Health | 1F32GM143850-01 | Amelie A Raz |
| National Institutes of Health | T32GM008216 | Gabriela S Vida |
| National Institutes of Health | 1F31HD108918-01 | Mara R Grace |

| Funder | Grant reference number | Author |
| --- | --- | --- |
| National Science Foundation | 2022305201 | Jasmine R Grey |
| National Science Foundation | 2021316576 | Jennifer M Viveiros |
| National Institutes of Health | T32GM007790 | Sarah R Stern |
| National Institutes of Health | T32AR007422 | Cameron W Berry |
| Howard Hughes Medical Institute | | Yukiko M Yamashita |
| Canadian Institutes of Health Research | PJT-133743 | Julie A Brill |
| Natural Sciences and Engineering Research Council of Canada | RGPIN-2016-06775 | Julie A Brill |
| Natural Sciences and Engineering Research Council of Canada | RGPIN-2022-05163 | Julie A Brill |
| National Institutes of Health | R35GM136665 | Erika L Matunis |
| National Institutes of Health | R01HD052937 | Erika L Matunis |
| National Institutes of Health | R35GM136433 | Margaret T Fuller |
| Stanford University | Reed-Hodgson Professorship in Human Biology | Margaret T Fuller |
| National Institutes of Health | R00AG062746 | Hongjie Li |
| Cancer Prevention and Research Institute of Texas | RR200063 | Hongjie Li |
| National Institutes of Health | R35GM136270 | Stephen DiNardo |
| Biotechnology and Biological Sciences Research Council | BB/P001564/1 | Helen White-Cooper |
| National Institutes of Health | NIDDK Intramural Research Program | Brian Oliver |
| Stanford University | Katherine Dexter McCormick and Stanley McCormick Memorial Professorship | Margaret T Fuller |

The funders had no role in study design, data collection and interpretation, or the decision to submit the work for publication.

## Author contributions

Amelie A Raz, Data curation, Software, Validation, Investigation, Visualization, Methodology, Writing – original draft, Writing – review and editing, Coordinated input from all authors; Gabriela S Vida, Software, Validation, Investigation, Visualization, Methodology, Writing – review and editing; Sarah R Stern, Jennifer M Viveiros, Mara R Grace, Software, Validation, Investigation, Visualization, Methodology, Writing – original draft, Writing – review and editing; Sharvani Mahadevaraju, Validation, Investigation, Visualization, Methodology, Writing – review and editing; Jaclyn M Fingerhut, Software, Formal analysis, Validation, Investigation, Visualization, Methodology, Writing – original draft, Writing – review and editing; Soumitra Pal, Data curation, Software, Validation, Visualization,

Methodology, Writing – review and editing; Jasmine R Grey, Software, Investigation, Visualization, Methodology, Writing – original draft, Writing – review and editing; Cameron W Berry, Software, Validation, Investigation; Hongjie Li, Investigation, Methodology; Jasper Janssens, Software, Visualization; Wouter Saelens, Software, Methodology; Zhantao Shao, Chun Hu, Investigation; Yukiko M Yamashita, Conceptualization, Resources, Supervision, Funding acquisition, Project administration; Teresa Przytycka, Software, Supervision, Funding acquisition; Brian Oliver, Conceptualization, Resources, Supervision, Funding acquisition, Validation, Project administration; Julie A Brill, Supervision, Funding acquisition, Investigation, Project administration, Writing – review and editing; Henry Krause, Supervision, Funding acquisition, Investigation, Visualization, Project administration; Erika L Matunis, Conceptualization, Resources, Supervision, Funding acquisition, Validation, Visualization, Project administration, Writing – review and editing; Helen White-Cooper, Conceptualization, Supervision, Funding acquisition, Validation, Investigation, Visualization, Project administration, Writing – review and editing; Stephen DiNardo, Conceptualization, Resources, Software, Supervision, Funding acquisition, Visualization, Writing – original draft, Project administration, Writing – review and editing; Margaret T Fuller, Conceptualization, Resources, Supervision, Funding acquisition, Investigation, Visualization, Writing – original draft, Project administration, Writing – review and editing

**Author ORCIDs**
Amelie A Raz http://orcid.org/0000-0002-4651-0626
Sarah R Stern http://orcid.org/0000-0002-6518-7289
Jaclyn M Fingerhut http://orcid.org/0000-0002-2347-0799
Soumitra Pal http://orcid.org/0000-0003-4840-3944
Jasmine R Grey http://orcid.org/0000-0001-6095-4461
Julie A Brill http://orcid.org/0000-0002-5925-9901
Stephen DiNardo http://orcid.org/0000-0003-4131-5511
Margaret T Fuller http://orcid.org/0000-0002-3804-4987

**Decision letter and Author response**
Decision letter https://doi.org/10.7554/eLife.82201.sa1
Author response https://doi.org/10.7554/eLife.82201.sa2

## Additional files

### Supplementary files
• MDAR checklist

### Data availability
Scripts written in the programming language R, necessary to reproduce the analyses in this manuscript are contained in a folder hosted at: https://doi.org/10.5061/dryad.m63xsj454. All additional data files, including the Seurat object for the snRNA-seq data (FCAloomToSeurat2TFP_Annotations.rds), the Monocle3 object for the scRNA-seq data (single_cell_monocle_obj.Robj), files for the warped pseudotime data, the full transcription factor list, and the 3D projection of the cyst cell lineage, as well as the original mapped gene expression matrix for the scRNA-seq data, are available at https://doi.org/10.5061/dryad.m63xsj454. Both the raw and the filtered data from the FCA snRNA-seq analysis are publicly available as indicated in the Results section of the text. The raw and processed scRNAseq data can be found at the GEO accession GSE220615.

The following datasets were generated:

| Author(s) | Year | Dataset title | Dataset URL | Database and Identifier |
|---|---|---|---|---|
| DiNardo S, Fuller M, Raz A | 2023 | Data for: Emergent dynamics of adult stem cell lineages from single nucleus and single cell RNA-Seq of *Drosophila* testes | https://dx.doi.org/10.5061/dryad.m63xsj454 | Dryad Digital Repository, 10.5061/dryad.m63xsj454 |
| Raz AA, Yamashita Y, Fuller M, Dinardo S | 2023 | Emergent dynamics of adult stem cell lineages from single cell RNA-Seq of *Drosophila* testes | https://www.ncbi.nlm.nih.gov/geo/query/acc.cgi?acc=GSE220615 | NCBI Gene Expression Omnibus, GSE220615 |

The following previously published dataset was used:

| Author(s) | Year | Dataset title | Dataset URL | Database and Identifier |
|---|---|---|---|---|
| De Waegeneer M, Janssens J, Li H, Aerts S | 2021 | The Fly Cell Atlas: single-cell transcriptomes of the entire adult *Drosophila* - 10x dataset | https://www.ebi.ac.uk/biostudies/arrayexpress/studies/E-MTAB-10519 | ArrayExpress, E-MTAB-10519 |

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
