## [Editor Report]

This paper uses single-nucleus RNA-seq and new single-cell RNA-seq data to provide an extensive characterization of cell types found within the *Drosophila* testis. This work provides a detailed analysis of the developmental transitions experienced by cells in the germline stem cell and cyst stem cell lineages. Comparison of snRNA-seq and scRNA-seq also reveals differences in steady-state RNA levels and ongoing transcription, providing insights into how gene expression patterns change during the development of these different cell populations. This study provides an important roadmap for future work using *Drosophila* testes to study development, reproduction, and stem cell biology.

---

## [Decision Letter]

**Decision letter after peer review:**

Thank you for submitting your article "Emergent dynamics of adult stem cell lineages from single nucleus and single cell RNA-Seq of *Drosophila* testes" for consideration by *eLife*. Your article has been reviewed by 4 peer reviewers, including Michael Buszczak as the Reviewing Editor and Reviewer #1, and the evaluation has been overseen by Utpal Banerjee as the Senior Editor. The following individual involved in review of your submission has agreed to reveal their identity: Todd G Nystul (Reviewer #3).

In general, the reviewers are enthusiastic about this manuscript. They agree that the work will have a positive impact on the field and is appropriate for publication in *eLife*. However, the reviewers collectively raised several important points that need to be addressed before publication.

Essential revisions:

1) The authors should provide more characterization and experimental validation of the bifurcation of the cyst cells.

2) The authors should attempt to identify biologically meaningful stages of germ cell development by adjusting the clustering methods and testing additional predicted new markers.

3) The authors need to address the issue with heterogeneity in the hub cell cluster and with vasa expression outside the germline, either experimentally or with new analysis and expanded discussion.

4) The authors provide concrete evidence that scRNAseq and snRNAseq are not identical, reflecting differences between stored RNAs and active transcription. The authors should rewrite the section on comparisons between their scRNAseq to snRNAseq datasets to better emphasize the significance and novelty of their findings, even beyond the characterization of the *Drosophila* testis.

The authors are also encouraged to address each of the additional comments raised in the individual critiques included below. We look forward to seeing your revised manuscript.

*Reviewer #1 (Recommendations for the authors):*

The authors should make sure all the links to data and analysis files are correct.

Scale bars are missing in Figure 2B and 4F.

Label missing on X-axis in 5H

*Reviewer #3 (Recommendations for the authors):*

I am highly supportive of the publication of this study. However, I have a few comments for discussion with the other reviewers and consideration by the authors:

1. The differences in transcriptional profiles observed in single cell and single nucleus sequencing is very interesting and quite novel (at least, I haven't seen that anywhere else before). If possible, it would be nice to see if this can be validated experimentally. Are there one or more genes, for which the transcript can be observed by FISH to shift from an enrichment in the nucleus to an enrichment in the cytoplasm?

2. More experimental evidence supporting the prediction that the lineage bifurcates into head and tail cyst cells would be helpful. They provide experimental data for two genes, shg and geko, that they present as markers of tail cyst cells and head cyst cells, respectively, but only geko seems to support their conclusions. The expression of shg in cluster 68 (one of the HCC clusters) does not look that different to me than in cluster 84 (a TCC cluster) and, in the image of shg FISH (Figure 6S), shg mRNA looks pretty abundant everywhere. I see a brighter signal is near the head (arrow) than the tail (arrowhead) but could that just be because the same amount of mRNA is compacted into a smaller area in the head region?

3. All of the links listed in the Code and Data Availability section led to a DOI Not Found page for me. Public access to these scripts and data should be verified before publication.

*Reviewer #4 (Recommendations for the authors):*

I would suggest providing the supplementary data files at a lower resolution as they were difficult to scroll through as PDF files, presumably due to file size issues.

I would appreciate a little better description of the Seurat data. For example, what can we infer from a spot having a colour value that equates to below zero? I am also unclear about what is represented by a value of zero (or near zero), yet 25% of cells show this value, e.g., in Figure 2L the CycB spot in cluster 37 – does this represent real expression of CycB in these cells or not, or is it expression at a level that would not be regarded as functional (although I am not sure how you would define this level).

In the paragraph beginning line 297 the authors state surprise that later stage spermatocytes split into three parallel streams. They indicate that nuclei in the leftmost and middle streams had considerably lower UMI counts than the mainstream and then go on to suggest a stochastic component to chromosome condensation, etc. Could it be that these streams are beginning the process of cell death? Is there a good measure of the percentage of spermatocyte cysts that die in a normal testis? Is there also any way to determine if these are entire cysts behaving in an aberrant manner or if they are single spermatocytes within a cyst where the rest of the cells are "mainstream"?

---

## [Author Response]

Essential revisions:1) The authors should provide more characterization and experimental validation of the bifurcation of the cyst cells.

We now provide significantly more characterization of the cyst cell subclusters in Figure 6, replacing several panels, and including coexpression enabling the distinction between HCCs and TCCs. We confirmed FISH analysis of *Nep4*, validating the existence of the split, and confirmed that *geko* expression was enriched to one arm. Co-labeling with a junctional marker strongly suggests that this arm represents the HCC cell type. Since we have not yet identified a gene reciprocally enriched to the other arm, in the revised submission we leave the assignment of TCC identity ‘tentative’. We end the section pointing out that genes predicted to be enriched to one or the other arm represent fertile candidates for the field to test.

2) The authors should attempt to identify biologically meaningful stages of germ cell development by adjusting the clustering methods and testing additional predicted new markers.

We have adjusted clustering methods, testing four predicted new markers and show that these indeed delineate discrete differentiation stages (new Figure 2 – —figure supplement 1).

3) The authors need to address the issue with heterogeneity in the hub cell cluster and with vasa expression outside the germline, either experimentally or with new analysis and expanded discussion.

We have analyzed *vasa* expression in more depth and now show that expression outside the germline appears quite minimal (new Figure 1 figure supplement panel). Whether or not this reflects minimal contamination by ambient RNA, *vasa* is observed in the single cell RNA-seq dataset also, so this does not reflect any limitation to the single nuclear RNA-seq. We also note that some of this somatic expression might in fact reflect the fact that *vasa* is indeed expressed in certain somatic cells (Renault, 2012).

We also clarified our analysis of the hub cell cluster, now adding several independent subclustering methods (both graph-based and hierarchical) to corroborate our annotation of the hub cells.

4) The authors provide concrete evidence that scRNAseq and snRNAseq are not identical, reflecting differences between stored RNAs and active transcription. The authors should rewrite the section on comparisons between their scRNAseq to snRNAseq datasets to better emphasize the significance and novelty of their findings, even beyond the characterization of the *Drosophila* testis.

We have re-written the results and Discussion sections regarding the comparison of sc- and sn-RNAseq to highlight the significance of this finding. In addition, in a new figure panel we provide a compelling example using double-label FISH with intronic vs exonic probes from the same gene showing cessation of transcription in the nucleus (intronic probe) but continued perdurance of cytoplasmic (exonic probe) signal through later differentiation stages, as predicted by our scRNA-seq to snRNA-seq comparison.

Reviewer #1 (Recommendations for the authors):The authors should make sure all the links to data and analysis files are correct.

Referees should receive an active link with the resubmission. For the initial submission, we supplied the *eLife* administrators with an active link, which they were to pass on to referees. We do not know why this did not happen.

Scale bars are missing in Figure 2B and 4F.

The scale bars have been added.

Label missing on X-axis in 5H

The axis label has been added.

Reviewer #3 (Recommendations for the authors):I am highly supportive of the publication of this study. However, I have a few comments for discussion with the other reviewers and consideration by the authors:1. The differences in transcriptional profiles observed in single cell and single nucleus sequencing is very interesting and quite novel (at least, I haven't seen that anywhere else before). If possible, it would be nice to see if this can be validated experimentally. Are there one or more genes, for which the transcript can be observed by FISH to shift from an enrichment in the nucleus to an enrichment in the cytoplasm?

We are pleased to report that we have added FISH data that demonstrate the predicted shift from active transcription in spermatocytes to stored cytoplasmic mRNA in spermatids.. We carried out double-label FISH using an intronic (nuclear) and an exonic (cytoplasmic) probe for *l(2)41Ab,* which is predicted from our sequencing data to be actively transcribed in early spermatocytes, to not be transcribed in early spermatids, but to be stored as perduring, cytoplasmic mRNA in early to late spermatids. In a new panel in Figure 4, we show by FISH that this is indeed the case: the nuclear intronic signal is detected in spermatocytes but not early spermatids, whereas signal from the exonic probe was detected in the cytoplasm through the spermatid stages. This exciting result indicates that the shift from nuclear to cytoplasmic transcript presence revealed by comparing sc- to snRNAseq data reflects real biology of transcript perdurance.

2. More experimental evidence supporting the prediction that the lineage bifurcates into head and tail cyst cells would be helpful. They provide experimental data for two genes, shg and geko, that they present as markers of tail cyst cells and head cyst cells, respectively, but only geko seems to support their conclusions. The expression of shg in cluster 68 (one of the HCC clusters) does not look that different to me than in cluster 84 (a TCC cluster) and, in the image of shg FISH (Figure 6S), shg mRNA looks pretty abundant everywhere. I see a brighter signal is near the head (arrow) than the tail (arrowhead) but could that just be because the same amount of mRNA is compacted into a smaller area in the head region?

We agree that the FISH analysis of *shg* expression did not make a conclusive argument that clusters 57, 80, 84, and 72 represent TCC nuclei, given the modest expression of *shg* in HCCs. Consequently, we have removed the *shg* panels from Figure 6. To clarify the identity of the bifurcation to the degree possible, we first confirmed FISH analysis of *Nep4* (Figure 6Q), which validates the existence of the split, and confirmed that *geko* expression was enriched to one arm (Figure 6T). As requested, we now present these panels at higher magnification, and we also co-labeled with Coracle, which demarcates the septate junction between HCC and TCC (Figure 6O, Q,S,T). These data strongly suggest that the arm of the bifurcation enriched for *geko* represents the HCC cell type. Since we have not yet identified a gene enriched reciprocally in the other arm, in the revised submission we leave the assignment of TCC identity ‘tentative’.

3. All of the links listed in the Code and Data Availability section led to a DOI Not Found page for me. Public access to these scripts and data should be verified before publication.

Referees should receive an active link with the resubmission. For the initial submission, we supplied the *eLife* administrators with an active link, which they were to pass on to referees. For the resubmission the proper link should be provided by the *eLife* administrators.

Reviewer #4 (Recommendations for the authors):I would suggest providing the supplementary data files at a lower resolution as they were difficult to scroll through as PDF files, presumably due to file size issues.I would appreciate a little better description of the Seurat data. For example, what can we infer from a spot having a colour value that equates to below zero? I am also unclear about what is represented by a value of zero (or near zero), yet 25% of cells show this value, e.g., in Figure 2L the CycB spot in cluster 37 – does this represent real expression of CycB in these cells or not, or is it expression at a level that would not be regarded as functional (although I am not sure how you would define this level).

This is a good point. The Dot Plot expression values use scaled data, with the mean expression set at 0. Therefore, in the cases where Dot Plots show negative values, these indicate average expression in the group of cells of interest is below the average expression across the entire UMAP. This becomes relevant in Figure 3I and our discussion of low-level expression of somatic genes in spermatocytes.

In the paragraph beginning line 297 the authors state surprise that later stage spermatocytes split into three parallel streams. They indicate that nuclei in the leftmost and middle streams had considerably lower UMI counts than the mainstream and then go on to suggest a stochastic component to chromosome condensation, etc. Could it be that these streams are beginning the process of cell death? Is there a good measure of the percentage of spermatocyte cysts that die in a normal testis? Is there also any way to determine if these are entire cysts behaving in an aberrant manner or if they are single spermatocytes within a cyst where the rest of the cells are "mainstream"?

These are interesting hypotheses; anecdotally we do not see spermatocyte cysts dying at this frequency and we are interested in seeing whether we can detect this lower UMI in vivo.